# Copy number variation-associated lncRNAs may contribute to the etiologies of congenital heart disease

Yibo Lu[1,4], Qing Fang[1,4], Ming Qi[1,4], Xiaoliang Li[2], Xingyu Zhang[1], Yuwan Lin[1], Ying Xiang [1,3✉], Qihua Fu [1,3✉] & Bo Wang [1,3✉]

Copy number variations (CNVs) have long been recognized as pathogenic factors for congenital heart disease (CHD). Few CHD associated CNVs could be interpreted as dosage effect due to disruption of coding sequences. Emerging evidences have highlighted the regulatory roles of long noncoding RNAs (lncRNAs) in cardiac development. Whereas it remains unexplored whether lncRNAs within CNVs (CNV-lncRNAs) could contribute to the etiology of CHD associated CNVs. Here we constructed coexpression networks involving CNV-lncRNAs within CHD associated CNVs and protein coding genes using the human organ developmental transcriptomic data, and showed that CNV-lncRNAs within 10 of the non-syndromic CHD associated CNVs clustered in the most significant heart correlated module, and had highly correlated coexpression with multiple key CHD genes. *HSALNG0104472* within 15q11.2 region was identified as a hub CNV-lncRNA with heart-biased expression and validated experimentally. Our results indicated that *HSALNG0104472* should be a main effector responsible for cardiac defects of 15q11.2 deletion through regulating cardiomyocytes differentiation. Our findings suggested that CNV-lncRNAs could potentially contribute to the pathologies of a maximum proportion of 68.4% (13/19) of non-syndromic CHD associated CNVs. These results indicated that explaining the pathogenesis of CHD associated CNVs should take account of the noncoding regions.

[1] Pediatric Translational Medicine Institute, Shanghai Children's Medical Center, School of Medicine, Shanghai Jiao Tong University, Shanghai, China. [2] Department of Medical Genetics and Molecular Diagnostic Laboratory, Shanghai Children's Medical Center, School of Medicine, Shanghai Jiao Tong University, Shanghai, China. [3] Shanghai Key Laboratory of Clinical Molecular Diagnostics for Pediatrics, Shanghai, China. [4] These authors contributed equally: Yibo Lu, Qing Fang, Ming Qi. ✉email: 1262975038@qq.com; qfu@shsmu.edu.cn; booew@163.com

Congenital heart disease (CHD) is the most common congenital anomaly worldwide, with an incidence of 6~13 in 1000 live births[1–3]. Despite advances in surgical correction and clinical care guaranteed most patients surviving to adults, CHD remains a major cause of new born-related mortality. The etiologies of CHD were multifactional. To date, ~20–30% of the CHD cases could be identified with environmental or genetic factors, although such number might change with the wide application of new testing methods such as next generation sequencing (NGS)[4,5]. The largest next generation sequencing study of CHD cohort indicated that genetic etiologies could be identified for 1/3 of the patients with CHD; de novo variants (DNVs) and inherited autosomal recessive variants account for 8% and 2% of the patients respectively[6]. Copy number variations (CNVs) are important sources of genetic etiologies of CHD. Pathogenic CNVs were recorded in 3–25% of syndromic CHD cases and 3–10% of isolated CHD cases[4]. The pathogenicity of CNVs involving coding sequences were usually interpreted based on their effect on gene dosage. Despite the success of such strategy in many diseases, the pathologies of most CHD associated CNVs remained undetermined[4,5,7].

Organogenesis of the heart is a complex process involving differentiation, specification, migration of multiple cell lineages, which requires elaborate gene regulatory networks initialized and governed by core lineage determining transcription factors (TFs) including NKX2-5, MESP1, GATA4, GATA6, and TBX5[5]. Over the last two decades, many studies have revealed that a large fraction of the noncoding genome (primary transcripts and processed transcripts cover 74.7% and 62.1% of the human genome respectively) was transcribed[8]. Long noncoding RNAs (lncRNAs), which are defined as transcripts of greater than 200 nucleotides without coding potential, emerge as key components in gene regulatory networks controlling cell fates during development[8,9]. Since lncRNA Braveheart (Bhvt) associated with cardiovascular development was discovered in mouse[10], dozens of lncRNA such as Fendrr[11], Chast[12], HBL1[13], Uph[14], Hdn[15], BANCR[16], and lncExACT1[17] have been found to be involved in cardiac developmental processes in cell and animal models. CHD is characterized by its high genetic heterogeneity, which made the discovery of pathogenetic lncRNAs frustratingly difficult. Yet benefited from the accumulated evidences that established the robust association of CNVs with CHD, recurrent pathogenic CNVs have provided a natural source to link lncRNAs to disease phenotypes.

Most CNVs affect genomic regions encompassing lncRNAs. Contribution of lncRNAs in CNV-related pathogenesis have been implicated in several neurological studies. Meng et al. investigated lncRNAs within 10 schizophrenia risk-associated CNV regions and identified DGCR5 within 22q11.2 as a hub gene regulating schizophrenia-related genes[18]. Alinejad-Rokny et al. identified 47 recurrent autism spectrum disorder associated CNVs and showed that brain-enriched coding genes and lncRNAs were over-represented in these regions[19]. Recently, a trio-based whole genome sequencing study of 749 CHD probands demonstrated an enrichment of potentially disruptive regulatory noncoding de novo variants (DNVs)[20], which emphasized the importance of noncoding variations in CHD study. Meerschaut et al. performed retrospective reassessment of 138 CNVs with unknown pathogenicity and proposed potential relevance of non-coding gene regulatory elements in CNV-related CHD pathogenesis[21]. Whereas, the contributions of lncRNAs located in CNV regions (CNV-lncRNAs) to the etiologies of CHD have not been systematically evaluated.

We hypothesized that CHD associated CNVs might disrupt some, if not all, of the CNV-lncRNAs, which would consequently dysregulate their target genes and contribute to the etiologies of CHD. To test our hypothesis, we summarized recurrent CHD associated CNVs and retrieved candidate CNV-lncRNAs located within these regions. Integrated coexpression profile of such CNV-lncRNAs and protein coding genes was built based on weighted gene coexpression network analysis (WGCNA)[22] of human organ developmental transcriptomic data from LncExpDB[23]. We identified two modules significantly correlated with heart tissues, one of which showed enrichment of known CHD genes. It was noticeable that CNV-lncRNAs from 52.6% (10/19) of all non-syndromic CHD associated CNVs clustered in the most significant heart module. The hub CNV-lncRNA HSALNG0104472 of this module was located in the 15q11.2 region, deletion of which was previously proved to be associated with total anomalous pulmonary venous connection (TAPVC)[24]. We then conducted in vitro experiments to validate the potential regulatory effect of HSALNG0104472 on the known CHD genes and highlighted its potential role in cardiac development (Fig. 1).

## Results

**Recurrent CHD associated CNVs and CNV-lncRNAs.** Totally 19 CNVs including two deletions, six duplications and 11 deletions/duplications were defined as recurrent non-syndromic CHD associated CNVs (Fig. 2a; Table 1; Supplementary Fig. 1a; and Supplementary Table 1)[25–30]. A total of 568 candidate CNV-lncRNAs meeting the criterion together with 19957 protein coding genes were used for further investigation (Supplementary Data 1). We also extended our analysis to 21 syndromic CHD associated CNVs[4] (Supplementary Table 2). In comparison, most syndromic CHD associated CNVs (19/21) were deletions, and the other 2 CNVs could be either deletion or duplication (Supplementary Fig. 1b; Supplementary Table 2). Four CNVs (1q21.1 deletion/duplication, 8p23.1 deletion/duplication, 16p12.2 deletion/duplication, and 22q11.21 deletion/duplication) were associated with both non-syndromic and syndromic CHD cases (Fig. 2a).

**Construction of coexpression modules with non-syndromic CHD associated CNV-lncRNAs.** WGCNA was performed on the human organ developmental transcriptomic data from LncExpDB to construct coexpression modules involving candidate non-syndromic CHD associated CNV-lncRNAs and protein coding genes. We identified a total of 43 coexpression modules, 34 of which contained 511 of the 568 candidate CNV-lncRNAs. The remaining 57 candidate CNV-lncRNAs didn't cluster into any coexpression module (Supplementary Data 1). Additionally, 13 modules contained at least one hub CNV-lncRNA with module membership (MM) ≥ 0.8 and $P < 0.05$ (Supplementary Data 1). We then computed the Pearson correlation coefficients between the modules and heart tissues (Quantized trait information of samples are listed in Supplementary Data 1), and revealed that two modules (black and darkgreen) showed significant positive correlation ($r > 0.6$, $P < 0.05$; Fig. 2b).

Protein coding genes in two heart-correlated coexpression modules were used to implement functional enrichment analyses with Gene Ontology (GO) and Kyoto Encyclopedia of Genes and Genomes (KEGG) pathways. Noticeably, the black module, which was most significantly correlated to heart (r = 0.88, $P = 3.20 \times 10^{-104}$), was related to muscle system process ($P_{adj} = 3.35 \times 10^{-47}$, $P_{adj}$ represents adjusted $P$ value), muscle contraction ($P_{adj} = 3.55 \times 10^{-42}$), heart process ($P_{adj} = 1.14 \times 10^{-33}$), heart contraction ($P_{adj} = 1.14 \times 10^{-33}$), and muscle tissue development ($P_{adj} = 3.04 \times 10^{-32}$) (Fig. 2c). These genes were also enriched in KEGG pathways of hypertrophic cardiomyopathy ($P_{adj} = 7.62 \times 10^{-13}$), dilated cardiomyopathy ($P_{adj} = 1.87 \times 10^{-12}$),

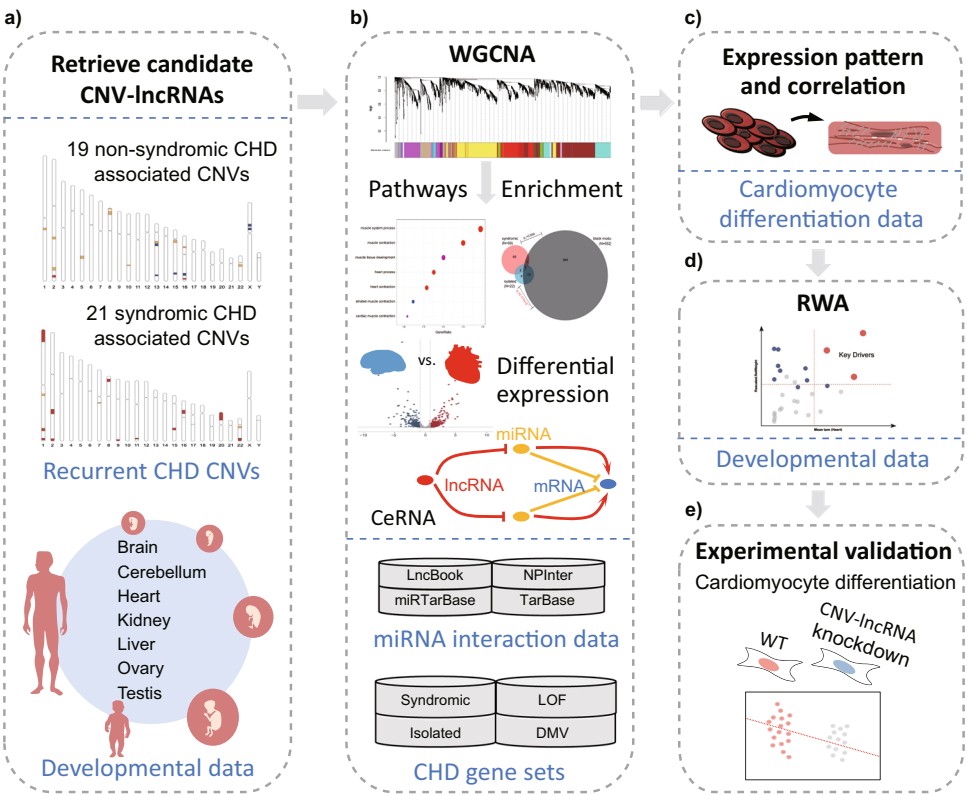

**Fig. 1 Study workflow. a** 19 recurrent non-syndromic CHD and 21 syndromic CHD associated CNVs were summarized. We retrieved candidate CNV-lncRNAs which were located within these regions and expressed during heart development based on human organ developmental transcriptomic data ($n = 313$) from LncExpDB. **b** Weighted gene coexpression network analysis (WGCNA) was performed to characterize the coexpression profile of CNV-lncRNAs and protein coding genes based on human organ developmental transcriptomic data. Downstream analyses including pathways analyses, enrichment analyses and differential expression analyses were performed to identify CHD-associated modules and hub CNV-lncRNAs. CNV-lncRNA-miRNA-mRNA regulatory networks were also identified based on miRNA interaction data and competing endogenous RNA (ceRNA) molecular mechanism. **c** Coexpression relationships in heart-associated non-syndromic black module were validated based on in vitro cardiomyocyte differentiation datasets ($n = 297$) from LncExpDB. **d** Relative weight analysis (RWA) revealed strong roles of hub CNV-lncRNA *HSALNG0104472* in regulating several key CHD genes. **e** In vitro experiments were performed to validate the predicted regulation of hub CNV-lncRNA *HSALNG0104472*.

cardiac muscle contraction ($P_{adj} = 1.15 \times 10^{-9}$), adrenergic signaling in cardiomyocytes ($P_{adj} = 4.24 \times 10^{-9}$), and arrhythmogenic right ventricular cardiomyopathy ($P_{adj} = 3.97 \times 10^{-7}$) (Supplementary Data 2). Mean expression values of CNV-lncRNAs in the heart developmental samples ($n = 50$) for the black module also ranked at the forefront of the 34 modules containing CNV-lncRNAs (ranked $3_{rd}$ of 34, mean tpm = 7.17; Supplementary Data 1). Based on the genomic locations relative to neighbor protein-coding genes, lncRNAs could be classified as intergenic, intronic (sense), intronic (antisense), overlapping (sense), overlapping (antisense), sense and antisense[31]. Sense, intergenic and antisense lncRNAs accounted for most of the CNV-lncRNAs (28/30) in the black module (Fig. 2d). Sequence conservation of 30 CNV-lncRNAs in the black module were identified according to alignment data from LncBook v2.0[32] (Fig. 2e; Supplementary Data 3). A high proportion of CNV-lncRNAs contained in the black module were highly correlated with multiple well characterized CHD genes such as *HAND1*, *HAND2*, *NKX2-5*, *TBX5*, *GATA6*, and *MYH6* (Fig. 3a). These lncRNAs distributed in 52.6% (10/19) of the recurrent non-syndromic CHD associated CNVs (Fig. 3a). Besides, correlations of the modules with developmental stage and sex were also computed (Supplementary Fig. 2 and Supplementary Data 1, 2). According to the genomic locus of CNV-lncRNAs and specific non-syndromic CHD associated CNV records, which included patients' phenotype, we identified 34 potential associations between CNV-lncRNAs and the CHD subtypes in the black module (Fig. 3b and Supplementary Data 4).

**Enrichment of known CHD genes in the heart associated coexpression modules**. To investigate the potential association between heart-correlated coexpression modules and CHD, we performed hypergeometric tests for enrichment analyses of these modules against four CHD-related gene lists (Supplementary Data 5): a 22-gene set responsible for monogenic causes of isolated CHD, a 69-gene set associated with monogenic conditions with syndromic CHD[5], a 66-gene set with loss of function (LOF) variants for CHD, and a 80-gene set with damaging missense variants (DMVs) for CHD[27]. The results indicated that two modules (black: $P = 2.12 \times 10^{-11}$, and salmon: $P = 0.04$) were enriched for the isolated CHD genes; two modules (turquoise: $P = 3.97 \times 10^{-6}$, and white: $P = 0.03$) were enriched for the syndromic CHD genes; three modules (pink: $P = 2.94 \times 10^{-5}$, black: $P = 1.34 \times 10^{-3}$, and turquoise: $P = 6.27 \times 10^{-3}$) were enriched for the LOF CHD genes, and five modules (black: $P = 5.62 \times 10^{-5}$, pink: $P = 3.02 \times 10^{-3}$, turquoise: $P = 5.35 \times 10^{-3}$, orange: $P = 6.78 \times 10^{-3}$, and darkorange: $P = 0.05$) were enriched for DMV CHD genes (Supplementary Data 5). It is particularly of interest that the black and turquoise modules significantly enriched three of the four CHD gene sets respectively (Fig. 4a–d and Supplementary Data 5). In addition, we found that a certain number of pathways, which were related to heart development, were significantly enriched in both black and turquoise modules, suggesting an important relationship between these modules and heart development (Fig. 4e and Supplementary Data 2).

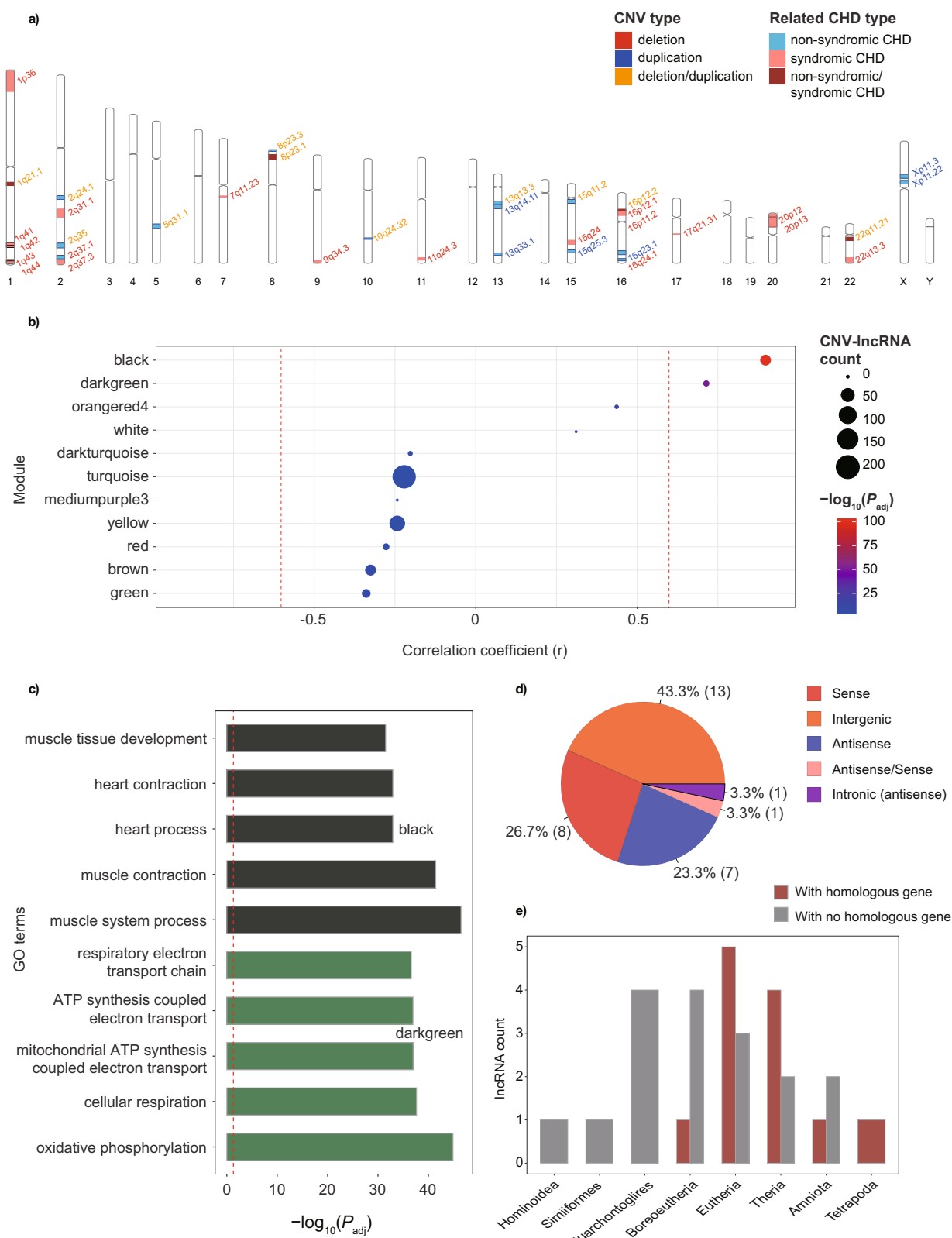

**Identification of CHD associated CNV-lncRNA-miRNA-mRNA regulatory network**. Competing endogenous RNA (ceRNA) has been recognized as an important molecular mechanism underlying mRNA expression regulated by lncRNA-miRNA interaction. Previously we successfully identified lncRNA-miRNA-mRNA regulatory networks in heart tissues of patients with tetralogy of Fallot[33] by implementing a causal inference method[34]. In the present study, we also applied this analysis to the human organ developmental transcriptomic data (n = 313). A total of 10 hub CNV-lncRNAs mediating ceRNA networks were identified (Supplementary Fig. 3; Supplementary Data 6). These CNV-lncRNAs distributed in the yellow module (*HSALNG0063477*), turquoise module (*HSALNG0112753*, *HSALNG0044817*, *HSALNG0006725*), grey60 module

**Fig. 2 Recurrent CHD associated CNV-lncRNAs in heart-correlated coexpression modules. a** Distribution of recurrent non-syndromic ($n = 19$) and syndromic CHD ($n = 21$) associated CNVs on human chromosomes are shown. The colors of the bands represent CHD case types in which CNVs were reported. The colors of the fonts represent CNV types. **b** Positively heart-correlated ($r > 0.6$, $n = 2$) coexpression modules (the black and darkgreen modules) constructed with human organ developmental dataset ($n = 313$) were identified (Supplementary Data 1). The y axis represents different CNV-lncRNA coexpression modules. Values of Pearson correlation coefficient ($r$) to heart tissue are shown on the x axis. The red dashed lines indicate $|r| = 0.6$. Sizes of the nodes represent CNV-lncRNAs count in each module. Colors of the nodes represent values of $-\log_{10}(P_{adj})$. Adjusted P value was caluculated with corPvalueStudent function in WGCNA R package. **c** Functional annotations of the positively heart-correlated coexpression modules are shown (Supplementary Data 2). Horizontal bars represent GO terms, and the colors of the bars represent different CNV-lncRNA coexpression modules. For each positively heart-correlated module, the top five GO terms (ranked by $P_{adj}$) are listed on the y axis. Values of $-\log_{10}(P_{adj})$ are shown on the x axis. The red dashed line indicates $P_{adj}$ of 0.05. **d** Classifications of CNV-lncRNAs in the black module. Counts for each class of the CNV-lncRNAs are shown in the parentheses. **e** Sequence conservation of the CNV-lncRNAs in the black module (Supplementary Data 3).

(*HSALNG0022874*), blue module (*HSALNG0110179*), and orange module (*HSALNG0022140*). *HSALNG0096627*, *HSALNG0019873*, *HSALNG0044901* did not cluster into any coexpression module and were labeled gray. Whereas, none of the known CHD genes was involved in these ceRNA networks. In addition, the CNV-lncRNAs and mRNAs, which shared a sponge lncRNA-mRNA regulatory relationship, rarely distributed in the same coexpression module (Supplementary Data 1 and 6).

**Extended and comparative analyses of syndromic and non-syndromic CHD associated CNV-lncRNAs.** CHD is frequently occurred as syndrome, and shared genetic contribution has been discovered for CHD and other anomalies, especially the neurodevelopmental disorders. We noticed that among the non-syndromic and syndromic CHD genes ($n = 88$; Supplementary Data 7), 13.6% (12/88) showed maximum expression in heart tissues. Surprisingly, 33.0% (29/88) showed maximum expression in brain and cerebellum tissues (brain: 19, cerebellum: 10, Supplementary Data 7). Differential expression analyses were performed for these CHD genes and involved CNV-lncRNAs between the heart ($n = 50$) and brain ($n = 87$) developmental samples (Supplementary Fig. 4). We found 21 heart-upregulated ($|log2FoldChange| \geq 1$, $P_{adj} < 0.05$) and 8 brain-upregulated genes in the known CHD gene list (Fig. 5a-c; Supplementary Data 7).

We further incorporated 21 syndromic CHD associated CNVs for comparative analysis (Supplementary Table 2). With the same criteria as previous analysis, 1500 CNV-lncRNAs were selected for WGCNA with the 19957 protein coding genes (Supplementary Data 1). Respective modules for syndromic and non-syndromic CHD were identified (Supplementary Data 8). We then tested the correlation between the syndromic modules and seven organs. The syndromic brown module ($r = 0.73$) with 57 CNV-lncRNAs and magenta module ($r = 0.71$) with 28 were highly correlated with brain, and the syndromic green module ($r = 0.87$) with 120 CNV-lncRNAs was highly correlated with cerebellum (Fig. 6 and Supplementary Data 8). The syndromic yellow module (s-yellow) should be corresponding to the non-syndromic black module: (1) The s-yellow module showed the highest correlations with heart tissue (r_heart = 0.83) (Figs. 2, 6); (2) Most (84.89%, 579/682) of the genes in the non-syndromic black module appeared in the s-yellow module (Supplementary Data 8); (3) Genes in two modules were enriched in the same functions (Supplementary Data 8). The gene types and sequence conservation of CNV-lncRNAs in s-yellow and s-turquoise modules were identified (Supplementary Figs. 5 and 6; and Supplementary Data 1 and 3). Differential expression analysis was also performed for 1500 CNV-lncRNAs between the heart and brain samples. We found 221 heart-upregulated ($|log2FoldChange| \geq 1$) and 205 brain-upregulated CNV-lncRNAs. (Fig. 5d and Supplementary Data 7). To further investigate the relationship between CHD and neurodevelopmental disorders, we described the distribution of the representative autism spectrum

disorder related genes[35] and tested their enrichment in the syndromic WGCNA modules. The results indicated that protein coding genes in the syndromic turquoise ($P = 7.66 \times 10^{-14}$), brown ($P = 9.18 \times 10^{-7}$), and mediumpurple3 ($P = 6.96 \times 10^{-4}$) modules were significantly enriched in autism spectrum disorder related genes (Supplementary Data 9). It was interesting to note that the non-syndromic turquoise (corresponding to s-turquoise) module significantly enriched both CHD and autism spectrum disorder gene sets (Fig. 4d and Supplementary Data 5 and 9). This module contained high proportion of brain-upregulated CNV-lncRNAs and syndromic CHD genes, which were related to multiple systems development and stem cell population maintenance (Fig. 5 and Supplementary Data 7).

***HSALNG0104472* may be a hub CNV-lncRNA regulating CHD associated genes in fetal cardiomyocyte.** According to above analyses, *HSALNG0104472* was identified as a hub CNV-lncRNA (module membership value of non-syndromic black module = 0.83; Supplementary Data 1) in the heart-correlated non-syndromic black module (Fig. 2b), and showed most biased expression in the heart samples compared with the brain samples (Fig. 5d and Supplementary Data 7). Relative weight analysis (RWA) also revealed its strong roles in regulating several key CHD genes (Fig. 7a and Supplementary Data 10). In addition, most of these coexpression relationships between *HSALNG0104472* and CHD genes were also observed in the induced pluripotent stem cells (iPSCs) based on in vitro cardiomyocyte differentiation datasets ($n = 297$) (Fig. 7b; Supplementary Fig. 7; and Supplementary Data 11).

Overexpression and knockdown of *HSALNG0104472* were performed in the adult human cardiomyocyte cell line (AC16). Our results suggested that knockdown of *HSALNG0104472* significantly affected the expression of 1310 protein coding genes ($|log2FoldChange| \geq 1$, $P_{adj} < 0.05$), which were enriched in pathways such as regulation of vasculature development and regulation of angiogenesis (Supplementary Data 12). Nevertheless, none of the CHD genes, which were predicted to be regulated by the CNV-lncRNAs, appeared in the differential expression gene list (Supplementary Data 12). Besides, overexpression of *HSALNG0104472* did not show significant effect on gene expression in AC16 cells (Supplementary Fig. 8).

**Reduction of *HSALNG0104472* significantly affects cardiomyocyte differentiation.** Successful shRNA-knockdown of *HSALNG0104472* was performed in iPSCs, and expression of *NKX2-5*, *ACTC1*, and *TBX20* were significantly downregulated (Supplementary Fig. 9a and Supplementary Data 13). Delayed occurrence of beating cardiomyocytes was observed in the in vitro induced *HSALNG0104472*-knockdown iPSCs (at day 9) compared with the control groups (at day 7) (Supplementary Fig. 10). The shRNA-knockdown of *HSALNG0104472* in iPSCs

**Table 1 Recurrent non-syndromic CHD associated CNVs.**

| CNV location | CNV type | Record count | Related CHD-risk genes | Associated CHD phenotype | Reference |
|---|---|---|---|---|---|
| 1q21.1 | Deletion/duplication | 20 | Unknown | TGA, ASD, MV, VSD, PDA, TOF, PA, CTD(TOF, APVS) | Soemediet al.[25] |
| 2q24.1 | Deletion/duplication | 3 | Unknown | TOF | Silversides et al.[26] |
| 2q35 | Deletion/duplication | 3 | Unknown | TOF, VSD | Silversides et al.[26] |
| 2q37.1 | Deletion | 3 | Unknown | LS-CHD, PA | Xie et al.[27] |
| 5q31.1 | Deletion/duplication | 3 | Unknown | TOF, TGA | Silversides et al.[26] |
| 8p23.3 | Deletion/duplication | 3 | Unknown | TOF, RLBV | Silversides et al.[26] |
| 8p23.1 | Deletion/duplication | 6 | GATA4 | TOF, ASD, DORV, CoA, LSVC, MA, VSD, CSD, CIAV, parachute mitral valve, transverse hypoplasia | Glessner et al.[28] |
| 10q24.32 | Deletion/duplication | 3 | Unknown | TGA, CoA | Sanchez-Castro et al.[29] |
| 13q33.1 | Duplication | 3 | Unknown | LS-CHD, PA, additional cardiac phenotype | Xie et al.[27] |
| 13q13.3 | Deletion/duplication | 3 | Unknown | TOF, VSD, PDA, PFO | Silversides et al.[26] |
| 13q14.11 | Duplication | 3 | Unknown | TOF, BAV, AI, ALV, TAPVC, VSD | Silversides et al.[26] |
| 15q11.2 | Deletion/duplication | 4 | Unknown | LVOT(CoA), CTD(TOF, PA), TAPVR | Glessner et al.[28] |
| 15q25.3 | Duplication | 3 | Unknown | CTD(TOF, DTGA), AS, CoA | Glessner et al.[28] |
| 16p12.2 | Deletion/duplication | 3 | Unknown | TGA, VSD, COA | Costain et al.[30] |
| 16q23.1 | Duplication | 3 | Unknown | ASD, PDA | Zhao et al.[37] |
| 16q24.1 | Deletion | 3 | Unknown | TOF, VSD | Silversides et al.[26] |
| 22q11.2 | Deletion/duplication | 53 | TBX1 | VSD, PS, ECD, TOF, PA, PDA, PA with MAPCA, APVS, IAA, DORV, CoA, PTA, AVS, HRH, RAA, BSVC, TA, AC, TGA, SA/SV, TAPVC, PAPVC, SAS, TVD, RVH, EA | Glessner et al.[28] |
| Xp11.22 (Xp11) | Duplication | 3 | Unknown | DOA, dilatation of aorta, LS-CHD, TOF | Silversides et al.[26] |
| Xp11.3 (Xp11) | Duplication | 1 | Unknown | EA | Sicko et al.[53] |

Note: 19 non-syndromic CHD associated CNVs that reported in at least three cases were listed (summarized from CHDGKB[39]). In order to narrow the subsequent screening scope of CNV-lncRNAs, the loci of 22q11, 22q11.2 were narrowed down to 22q11.21 because it has been clearly demonstrated that the mutation of TBX1 (located in 22q11.21) caused the pathogenicity of 22q11[54]. The region of Xp11 was also narrowed down to Xp11.22 and Xp11.3 according to the details of the case report (Supplementary Data 1). Full names of CHD phenotype were listed in Supplementary Table 1.

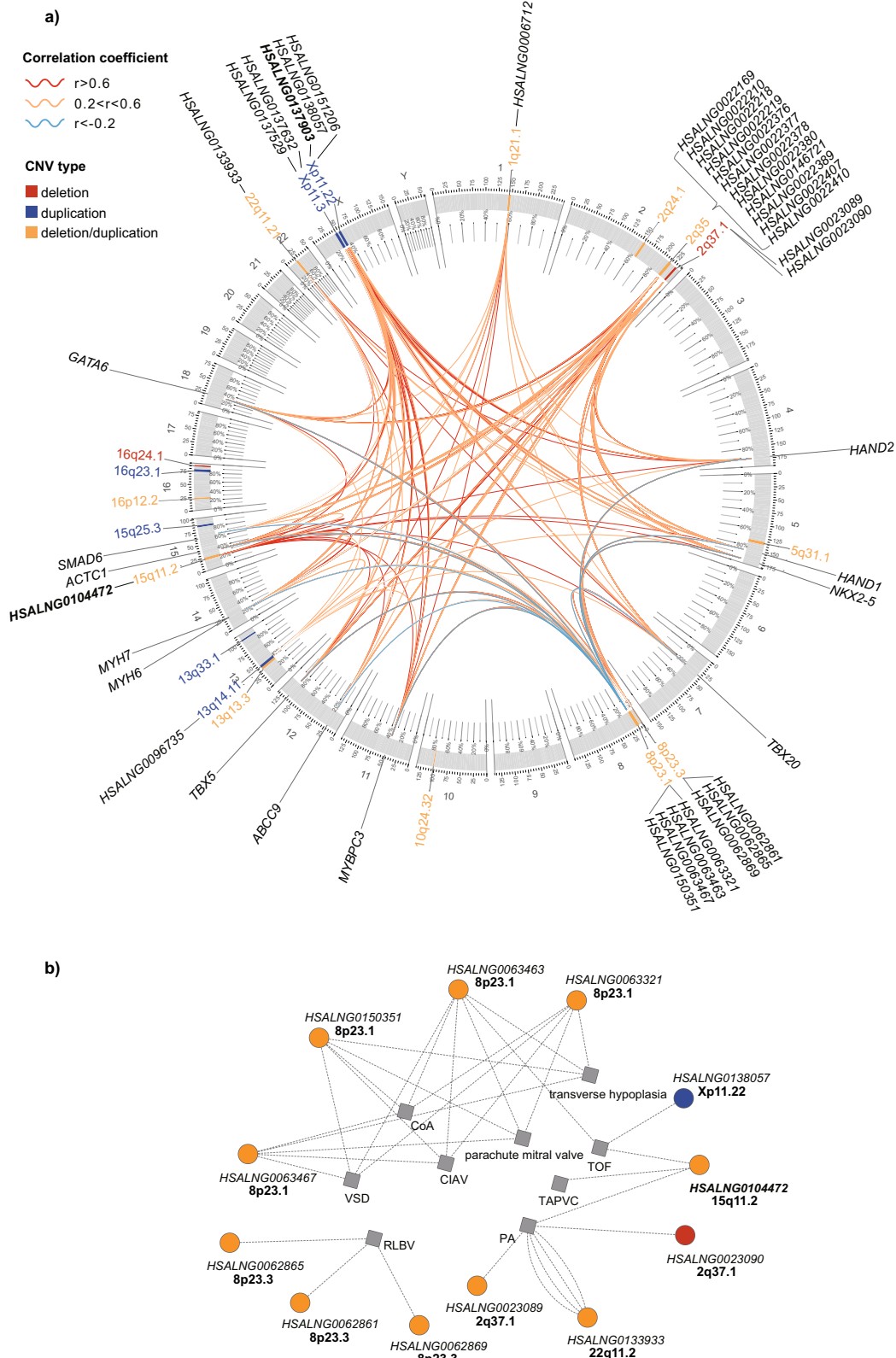

**Fig. 3 Black module contained co-expressed CNV-lncRNAs and CHD genes. a** Non-syndromic associated CNVs ($n = 19$), co-expressed CNV-lncRNAs ($n = 30$, distributing in 10 CNVs) and 12 CHD genes in heart-correlated black module are shown in the circos plot. The colors of CNVs represent CNV types. The colors of lines represent the Pearson correlation coefficient (calculated with human organ developmental dataset, $n = 313$) of each gene pair. Two hub CNV-lncRNAs in the black module are in bold font. **b** Correlations between CNV-lncRNAs and the CHD subtypes in the black module. These relationships are identified based on the intersection of CNV-lncRNAs and non-syndromic CHD associated CNVs (Supplementary Data 4).

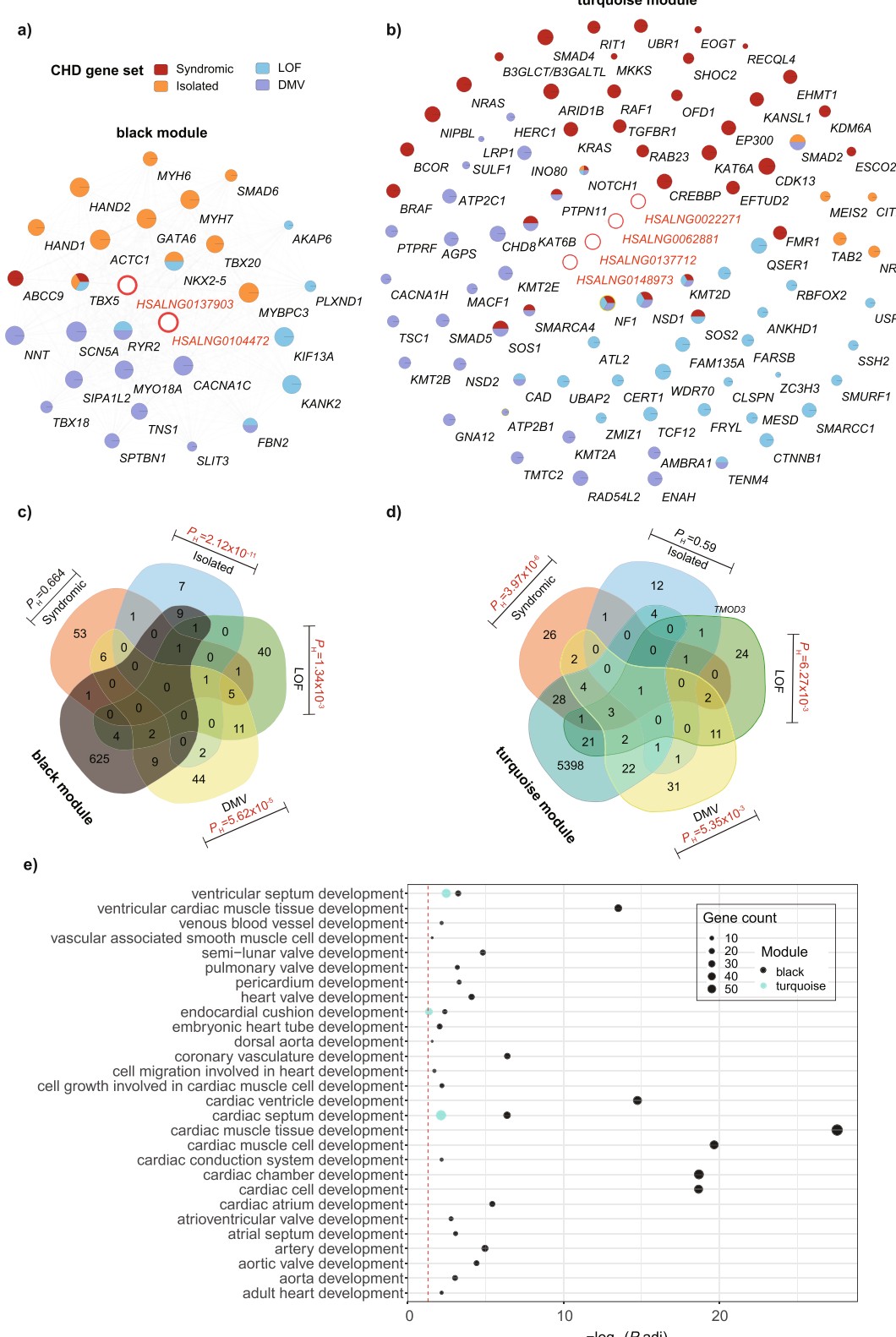

significantly affected the beating behavior of differentiated cardiomyocytes (at day 18). The beating frequency of knockdown group was 24 times per minute in contrast to 57 times per minute in control group. (Supplementary Movie 1 and 2). Significantly downregulated of *GATA6* was detected (Supplementary Fig. 9b and Supplementary Data 13). The *HSALNG0104472*-knockdown effect on cardiomyocyte differentiation was further validated

through transduction of a Smart Silencer containing small interfering RNA and antisense oligonucleotides in iPSCs following induction of cardiomyocyte differentiation (Fig. 8a). Flow cytometric analysis using the cardiomyocyte marker cardiac Troponin T (cTnT) indicated that knockdown of *HSALNG0104472* significantly reduced the cardiomyocyte differentiation efficiency ($P = 0.006$; Fig. 8b, Supplementary

**Fig. 4 Enrichment of known CHD genes in the heart associated coexpression modules.** Two heart associated modules, black and turquoise, that significantly enriched three of four CHD gene sets are shown. Hub CNV-lncRNAs and CHD genes in black (**a**) and turquoise (**b**) coexpression modules are listed (Supplementary Data 1 and 5). Hub CNV-lncRNAs are highlighted with red edge. Subsets of CHD genes are indicated in different colors. Sizes of the nodes represent gene module membership value (MM) in corresponding module (Supplementary Data 5). Hypergeometric test was used to calculate statistical significance for the enrichment of coexpressed protein coding genes in black (**c**) and turquoise (**d**) module against four CHD gene sets. CHD gene sets are indicated in different colors. Significant enrichment (PH < 0.05, PH represents hypergeometric P value) was shown with the PH value in red font. Protein coding genes that went for WGCNA ($n = 19,957$) were used as the background gene list. **e** Cardiac development related pathways in the black and turquoise modules (Supplementary Data 2). Horizontal axis represents GO terms. The colors of the dots indicate different modules. The size of the dots represents gene counts of each module involved in corresponding GO terms. Values of $-\log_{10}(P_{adj})$ are shown on the x axis. The red dashed line indicates $P_{adj}$ of 0.05.

Figs. 11–16, and Supplementary Data 13). We also validated the *HSALNG0104472*-knockdown effect in the differentiated cardiomyocytes (at day 18) through transient transfection. Immunofluorescent assay of the cardiomyocytes indicated reduced cardiac troponin I (cTnI) and lack of mature myocardial sarcomere resulted from *HSALNG01*04472 reduction (Fig. 8c).

## Discussion

CNVs had been recognized as important causes of CHD for a long time. Since CNVs range widely in size and always show multiple phenotypic effects, identifying the relevant gene or critical interval of a specific pathogenic CNV for CHD remained to be challenging[4]. In clinical applications, a natural idea for interpreting the pathogenicity of CNVs would be gene dosage effects resulted from disruption of coding sequences. Unfortunately, up to now only 2 (22q11.2: *TBX1* and 8p23.1: *GATA4*) of the 19 nonsyndromic CNVs had been documented as containing a relevant protein coding gene responsible for cardiac defects (Table 1). Obviously, such a strategy with identifying dosage-sensitive protein coding genes would be far from enough for elucidating the etiologies of CHD caused by CNVs.

Protein coding regions occupy only ~2% of the human genome. In contrast, a total of 60~70% of the human genome could be transcribed[8]. Therefore, it would be reasonable to examine the effects of noncoding RNAs in the etiologies of CHD associated CNV. During the past several years, lncRNAs had been implicated participating in orchestrating gene expression in cardiac developmental processes. In the current study, we investigated the potential regulatory roles of CNV-lncRNAs and their contribution to the etiologies of CHD caused by CNVs. Through WGCNA we identified two heart-tissue correlated coexpression modules containing a total of 35 CNV-lncRNAs (Supplementary Data 1). Functional enrichment analyses indicated these modules were related to biological processes such as heart process, muscle development, and energy metabolism, which had been recognized as important components of cardiac developmental process (Fig. 2c). The non-syndromic black module, which showed the highest correlation with heart tissues, contained half (11/22, $P = 2.12 \times 10^{-11}$) of the well characterized non-syndromic CHD genes but only 2 of the 69 syndromic CHD genes ($P = 0.66$). Besides, the protein coding genes in the black module were also enriched in the LOF ($P = 1.34 \times 10^{-3}$) and DMV ($P = 5.62 \times 10^{-5}$) CHD gene sets (Fig. 4a, c and Supplementary Data 5), which were obtained according to large scale rare variant discoveries using next generation sequencing[36]. Taken together, these results indicated that the non-syndromic black module had a disposition to affect cardiac phenotypes. Furthermore, more than a half (52.63%, 10/19) of the recurrent non-syndromic CHD associated CNVs encompassed at least one lncRNA coexpressed with multiple key CHD genes in the non-syndromic black module (Fig. 3a).

The non-syndromic black module contained 2 hub CNV-lncRNAs (*HSALNG0104472* in 15q11.2 and *HSALNG0137903* in

Xp11.22) (Fig. 4a and Supplementary Data 1). Relative weight analysis (RWA) showed that *HSALNG0104472* had significantly driven effect on the regulation of expression of all 12 CHD genes coexpressed in the non-syndromic black module, and *HSALNG0137903* significantly drove the regulation of 10 coexpressed CHD genes except *MYH6* and *SMAD6* (Fig. 7a and Supplementary Data 10). The 15q11.2 deletion had been repeatedly reported as contributing to neurodevelopment defects. However, its association with CHD was controversial[37]. We recently revealed that 15q11.2 deletion was associated with TAPVC, a rare and severe form of CHD[24]. Whereas, the critical genes of 15q11.2 deletion responsible for cardiac defects remained unknown. Interestingly, of the 30 CNV-lncRNAs in the non-syndromic black module, *HSALNG0104472* showed most significantly biased expression in the developmental heart compared with the brain (Fig. 5d and Supplementary Data 7). We then performed overexpression and knockdown experiments in cardiomyocyte cell lines to investigate the regulatory effects of *HSALNG0104472*. Overexpression of *HSALNG0104472* did not give rise to significant disturbance of gene expression in AC16 cell lines. With the knockdown of *HSALNG0104472* in AC16, we identified downregulated genes involved in pathways such as positive regulation of angiogenesis, positive regulation of vasculature development, and regulation of vasculature development. Nevertheless, we could not unambiguously validate the coexpression relationship between *HSALNG0104472* and the CHD genes identified in the non-syndromic black module (Supplementary Data 12). We speculated that such inconsistence should be attributed to the facts that the coexpression modules were constructed based on developmental datasets, whereas AC16 represents an adult human cardiomyocyte cell line. Further validation using the iPSCs-cardiomyocyte differentiation system indicated stage-specific regulatory effects of *HSALNG0104472* on the CHD genes (*NKX2-5*, *ACTC1*, and *TBX20* in iPSCs and *GATA6* in differentiated cardiomyocytes, Supplementary Fig. 9). The 15q11.2 deletion region mainly encompasses 4 protein coding genes: *TUBGCP5*, *CYFIP1*, *NIPA1*, and *NIPA2*. In this study, our differential gene expression analysis between developmental heart and brain tissues indicated that *NIPA1* was preferentially expressed in developmental brain tissues, whereas the other three genes did not show significant differential expression (Supplementary Data 7). Since only *TUBGCP5* and *NIPA1* were reported to be expressed in fetal hearts, we previously created *TUBGCP5* knockout iPSCs and proved that reduction of *TUBGCP5* would affect cardiomyocyte differentiation[24]. Here, we proved that the CNV-lncRNA *HSALNG0104472* specifically express in developmental heart tissues and its reduction would generate more severe impact on cardiomyocyte differentiation. Therefore, *HSALNG0104472* should be a major effector for cardiac defects resulted from 15q11.2 deletion.

The recent discovery of numerous novel CHD genes benefited from applications of large-scale next generation sequencing on CHD cohorts. Since most evidences of these genes (as summarized in LOF and DMV gene sets, Supplementary Data 5) were related to rare variants, it was reasonable that these genes were

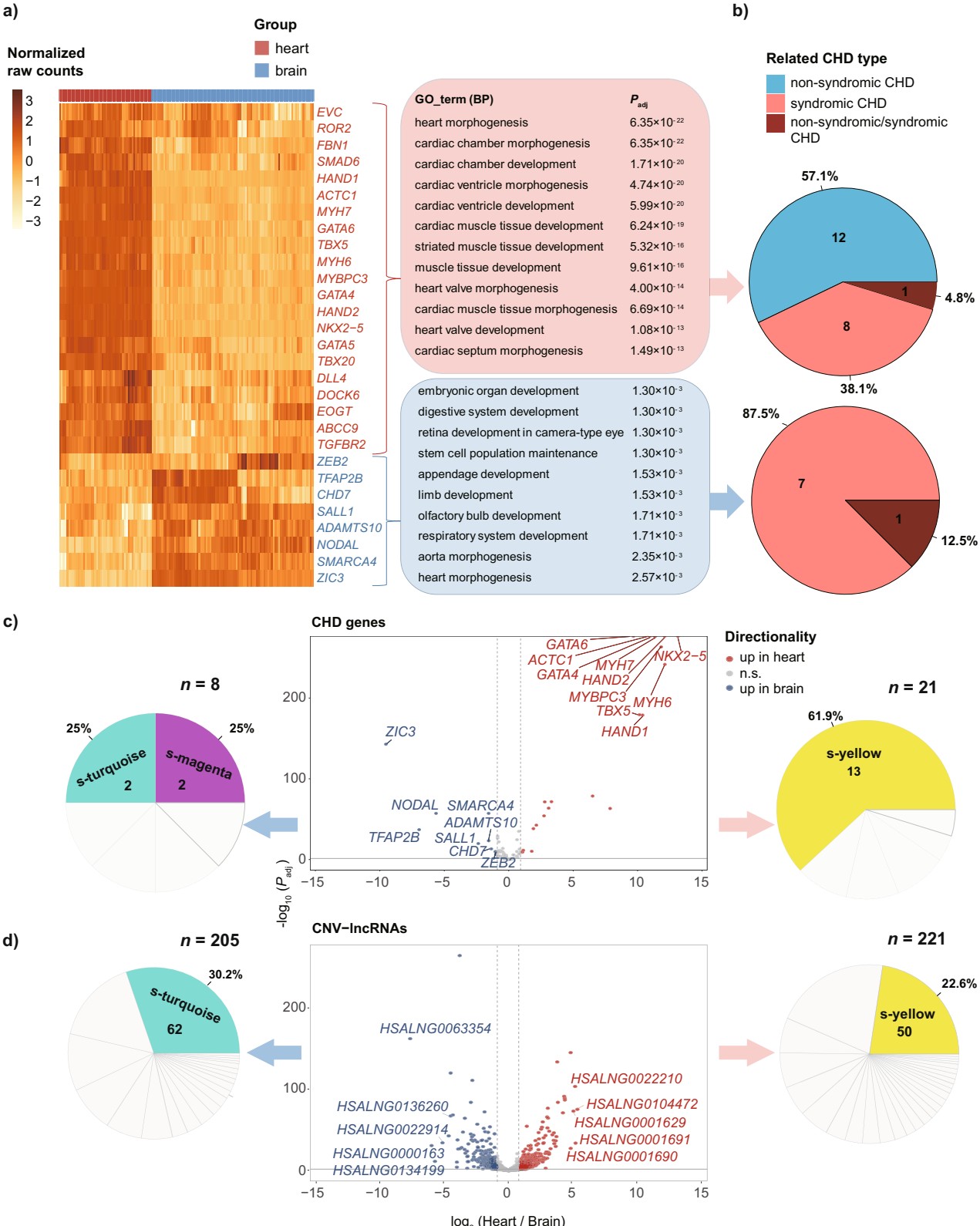

enriched in much more coexpression modules. Therefore, next generation sequencing-based rare variant discovery remarkably expanded our knowledge of the genetic etiologies of CHD. Such results were also consistent with the universal genetic heterogeneity of CHD. Syndromic and non-syndromic CHD gene sets were also enriched in different modules, suggesting divergence of molecular basis underlying syndromic and non-syndromic CHD

(Fig. 4 and Supplementary Data 5). Hence, we extended our analysis to lncRNAs within syndromic CHD associated CNVs. Compared with non-syndromic CHD, almost all syndromic CHD associated CNVs were deletions[4]. In general, deletions should be more deleterious than duplications[4]. On the other hand, a considerable number of recurrent deletion syndromes exhibited reduced penetrance or high clinical variability[7]. The syndromic

**Fig. 5 Differentially expressed CHD genes and CNV-lncRNAs in developmental heart and brain.** CHD genes upregulated ($|\log_2\text{FoldChange}| \geq 1$, $P_{adj} < 0.05$) in heart (21 genes) and brain (8 genes) developmental samples (Heart samples: $n = 50$, brain samples: $n = 87$) are shown in heatmap (**a**). For each cluster, statistically enriched GO Biological Process terms and $P_{adj}$ are shown on the panel. Pie plots show related CHD type of each differentially expressed CHD genes cluster (**b**). Top differentially expressed CHD genes (**c**) and CNV-lncRNAs (**d**) for each cluster are labeled in volcano plots (ranked by $\log_2\text{FoldChange}$). The x and y axes represent $\log_2\text{FoldChange}$ (heart vs brain) and $-\log_{10}(P_{adj})$, respectively. Red dots represent significantly upregulated genes in heart ($\log_2\text{FoldChange} \geq 1$, $P_{adj} < 0.05$). Blue dots represent significantly upregulated genes in brain ($\log_2\text{FoldChange} \leq -1$, $P_{adj} < 0.05$). Gray dots represent genes that do not differentially expressed. The horizontal and vertical red dashed line indicate $P_{adj} = 0.05$ and $|\log_2\text{FoldChange}| = 1$, respectively. Pie plots beside each cluster show distribution of genes in coexpression modules constructed by syndromic WGCNA. Only modules with highest gene proportion of each cluster are colored.

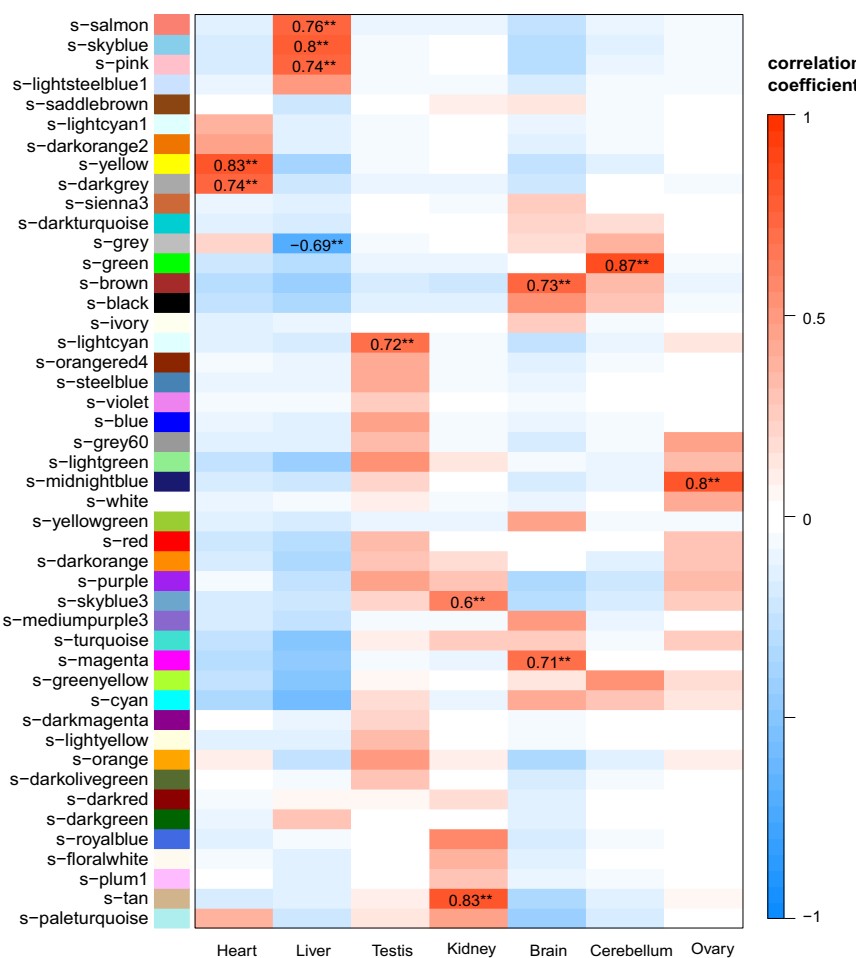

**Fig. 6 CNV-lncRNA coexpression modules related to seven organs.** Pearson correlation coefficient (*r*) between 45 coexpression modules and sample traits (Tissue types) were calculated in syndromic WGCNA (Supplementary Data 8). Only significant correlation ($|r| > 0.6$, $P_{adj} < 0.05$) are labeled. The colors represent correlation coefficient value and direction. \*\*$P_{adj} < 0.01$.

CHD associated CNVs contained more lncRNAs and hub CNV-lncRNAs, which were involved in more than twice (29/13) as many modules as non-syndromic CHD (Supplementary Data 1 and 8). Through differential gene expression analyses of the 88 syndromic and non-syndromic CHD genes and CNV-lncRNAs in heart and brain developmental samples, we identified 21 and 8 genes upregulated in the heart and brain respectively. The heart-upregulated genes were related to cardiac development, and the brain-upregulated genes were related to development of multiple systems (Fig. 5a–c). Strikingly, both the heart-upregulated protein coding genes and CNV-lncRNAs were most enriched in s-yellow module, which was corresponding to the non-syndromic black module (Fig. 5c, d and Supplementary Data 7). The largest turquoise module was enriched in the syndromic CHD, LOF, and DMV CHD gene sets (Fig. 4b, d and

Supplementary Data 5). We revealed that a large proportion of brain upregulated syndromic CHD associated CNV-lncRNAs were clustered in the corresponding (s-turquoise) module (Fig. 5d and Supplementary Data 7). Autism spectrum disorder represented a neurodevelopmental disorder with shared genetic basis of CHD. We tested autism spectrum disorder genes in the syndromic coexpression modules (Supplementary Data 9). In sum, these finding highlighted the importance of CNV-lncRNAs within the turquoise (s-turquoise) module in the etiologies of syndromic CHD with neurodevelopmental defects.

Environmental factors could be associated with up to 30% of CHD cases, whereas solitary environmental causes are identifiable in only 2% cases[4]. Most of the unexplained CHD cases were suggested to be caused by interactions of genetic and environmental factors, which might be modulated by epigenetic regulators.

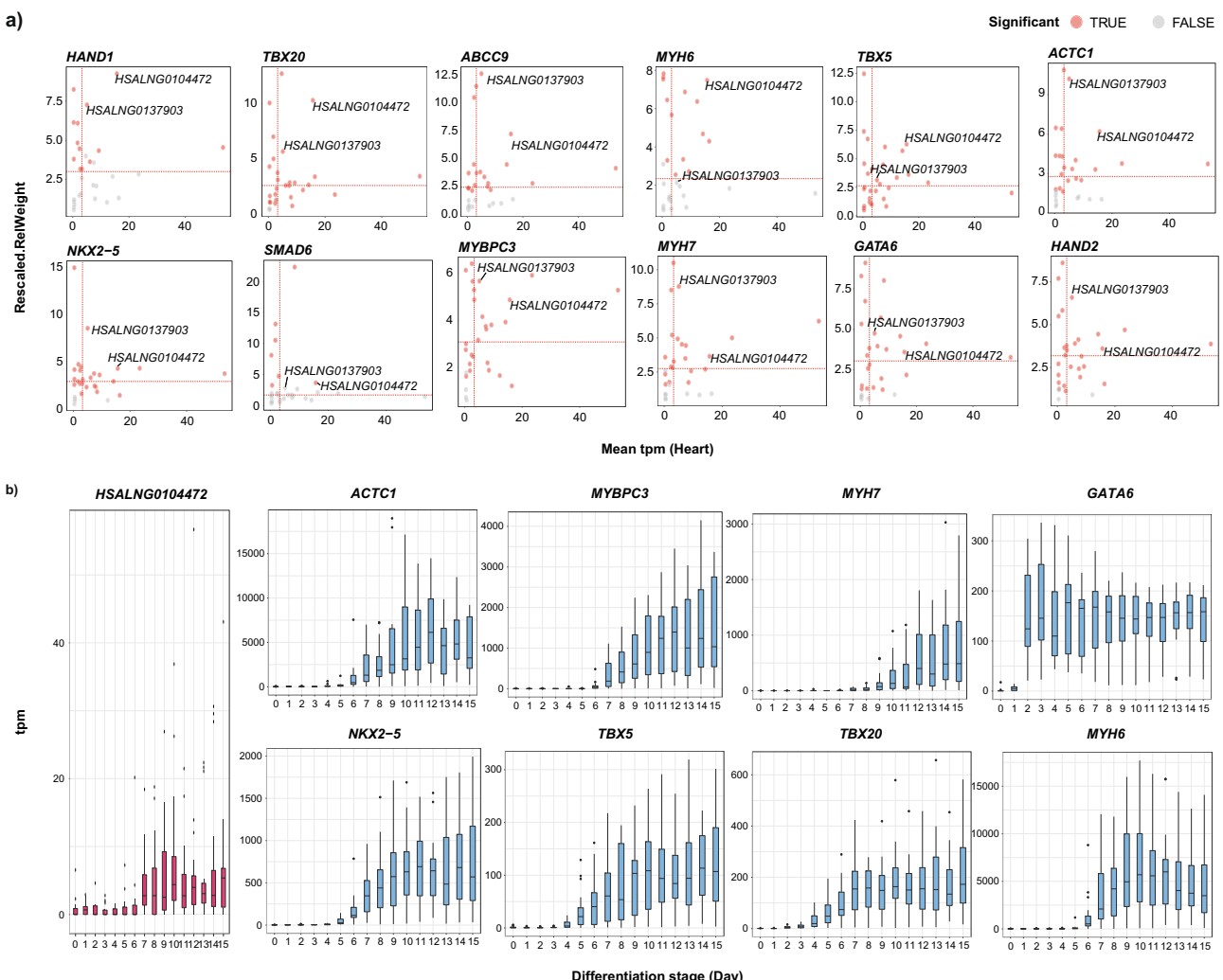

**Fig. 7 Regulatory effect of hub CNV-lncRNA *HSALNG0104472* predicted with datasets of developmental heart and in vitro differentiation from human iPSCs to cardiomyocytes. a** Relative weight of CNV-lncRNAs ($n = 30$) to key CHD genes in the non-syndromic black module are shown (Supplementary Data 10). Values of rescaled relative weight (as a percentage of predicted variance in the criterion variable attributed to each predictor, within rounding error rescaled weights of predictors in one test sum to 100%) are shown on the y axis, which represent the regulatory effect. Mean expression value (transcripts per million, tpm) of CNV-lncRNAs in the developmental heart samples ($n = 50$) are shown on the x axis. The red dashed lines indicate medium value of each axis. Red dots represent significant predictors. Gray dots represent nonsignificant predictors. **b** Expression patterns of *HSALNG0104472* and 8 predicted regulated CHD genes in the non-syndromic black module during in vitro differentiation from human iPSCs to cardiomyocytes ($n = 297$) (Supplementary Data 11). The x and y axes represent cardiomyocyte differentiation stage (day) and mean expression value (tpm) of each stage, respectively. The center line represents a median value. The box limits represent upper and lower quartiles. The whiskers represent 1.5x interquartile range. The points represent outliers.

Growing evidences for involvement of lncRNAs in cardiac development suggested that their dysregulation underlying CHD should be seriously considered. In contrast to mRNA and miRNAs, lncRNAs evolved rapidly from the perspective of sequence and expression levels. However, their tissue specificities are often conserved. LncRNAs could either repress or activate gene expressions in *cis* or in *trans*. According to our analyses, most CNV-lncRNAs were highly correlated with known CHD genes outside their corresponding CNV regions (Fig. 3a), thus functioning in *trans*. Mechanically, lncRNAs could be categorized as signaling lncRNAs, decoy lncRNAs, guide lncRNAs, scaffold lncRNAs, and enhancer lncRNAs. Previously, we successfully constructed lncRNA-miRNA-mRNA gene regulatory networks in heart tissues with CHD[33]. In present study, we also identified hub CNV-lncRNAs which might drive the lncRNA-miRNA-mRNA regulatory network (Supplementary Fig. 3). However, none of the target mRNAs were in the known CHD gene list (Supplementary Data 5).

The pathogenicity of CNVs was most commonly interpreted on the basis of their effect on gene dosage through disrupting coding sequences, and this approach has been successfully applied in causal gene discovery for CNVs in Mendelian diseases[7]. Despite the success of such strategy, the pathologies of most CHD associated CNVs have long remained undetermined. Since most CNVs also affect genomic regions of lncRNAs, pathogenic mechanisms involving regulatory lncRNAs need to be seriously considered. Surprisingly, even if we only considered the module (non-syndromic black module) that most significantly correlated with heart tissues, over a half of the non-syndromic CHD associated CNVs (52.6%, 10/19) contained at least one lncRNA showing high coexpression and correlation with multiple key CHD genes (Fig. 3a). Of the aforementioned 10 lncRNA-containing CNVs, 1q21.1, 2q24.1, 2q35, 2q37.1, 8p23.1, and 13q14.11 also encompassed hub lncRNA(s) involved in ceRNA mechanism. Apart from these CNVs, other three non-syndromic

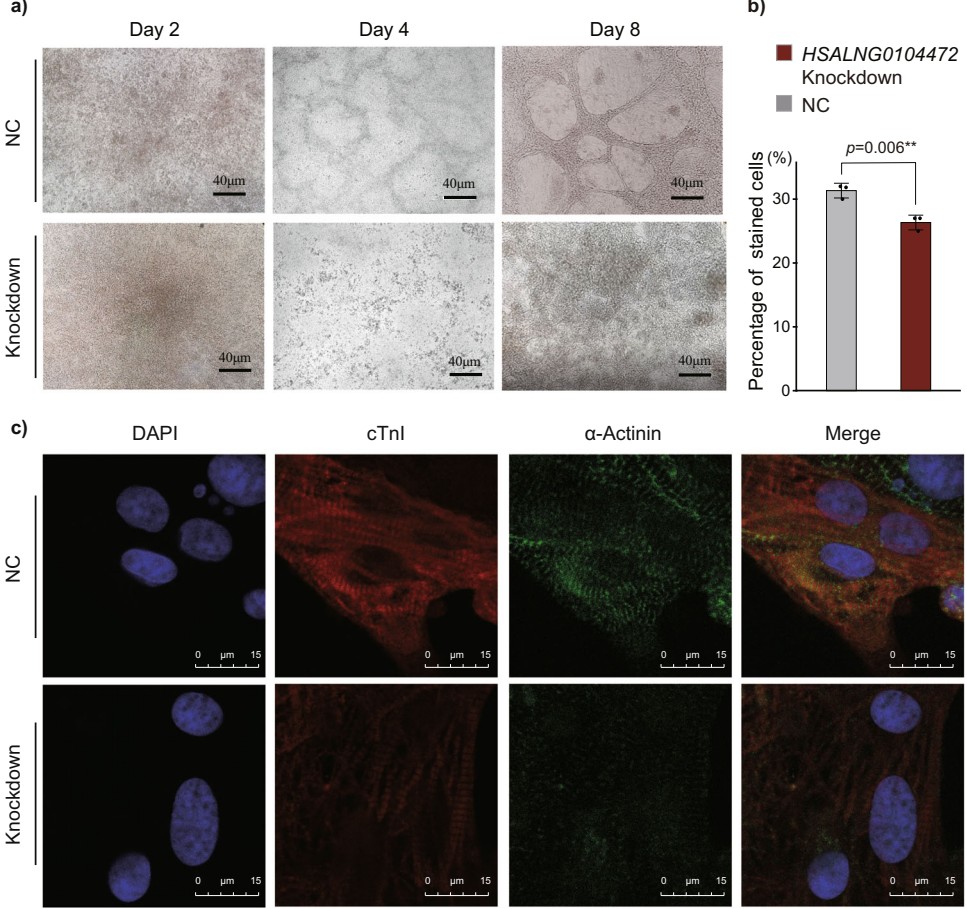

**Fig. 8 Reduction of CNV-lncRNA *HSALNG0104472* may affect cardiomyocyte differentiation. a** For *HSALNG0104472* knockdown and control groups, 3 time points during the differentiation of human iPSCs to cardiomyocytes were captured. Scale bar, 40 μm. **b** For *HSALNG0104472* knockdown and control groups, quantification of cardiomyocytes (at day 8 post induction) containing cardiac Troponin T (cTnT) is shown (Supplementary Figs. 11–16 and Supplementary Data 13). The error bars show mean ± SD of three biologically independent experiments. Two-tailed Student's *t* test was used for comparison between two group. **P < 0.01. **c** Immunofluorescence of cardiac sarcomere markers in induced cardiomyocytes. DAPI (blue), cTnI (red) and α-Actinin (green). Scale bar, 15 μm. cTnI cardiac troponin I.

CHD associated CNVs (5q31.1, 16p12.2, and 16q23.1) also contained hub lncRNAs implicated in the ceRNA analyses (Fig. 3a and Supplementary Data 6). Although CHD genes were not identified as targets of hub lncRNAs for the ceRNA network, we could not ignore the effect of ceRNA mechanism due to its ubiquitous regulatory roles in cardiac development[38]. Altogether, CNV-lncRNAs could potentially contribute to the pathologies of a maximum proportion of 68.4% (13/19) of the non-syndromic CHD associated CNVs (Fig. 3a; Table 1; and Supplementary Data 6).

Further investigations are needed since there are several limitations to our research. First, causal lncRNAs that could explain CHD cases resulted from CNVs should be identified from the coexpression networks, which was also the original motivation of this work. Second, we used four known CHD gene sets in the analyses (Supplementary Data 5). The results might be incomplete given the CHD gene list is still growing. In addition, although we grouped mRNA and CNV-lncRNAs into modules to simplify the analyses, the CNV-lncRNAs and mRNAs might have complex inter-module interactions in the temporal and spatial view. The hub CNV-lncRNA *HSALNG0104472*, for example, whose inconsistent regulatory effect on known CHD genes was observed in iPSC and differentiated cardiomyocytes (i.e., transcriptional levels of *NKX2.5*, *ACTC1*, and *TBX20* were down-regulated in iPSCs but not in differentiated cardiomyocytes),

indicated the stage-specific roles of *HSALNG0104472* (Supplementary Fig. 9 and Supplementary Data 13). Last, lack of deep mechanistic investigation is a limitation of the study. Altogether, we provided evidences that CNV-lncRNAs potentially regulated expression patterns of well-established CHD genes. Integration of multidimensional datasets are needed to reveal their molecular mechanisms in CHD. However, additional work is still necessary to fully reveal the regulatory roles of CNV-lncRNAs in CHD. Improvement in cell and in vitro/in vivo model plus accumulation of clinical genetic data should be needed to help us achieve such goal. Since most of the CNV-lncRNAs are not evolutionary conserved, in vitro models such as cardiac organoids have emerged as potential tools to provide insights of molecular mechanisms for them.

In conclusion, to investigate whether lncRNAs contribute to the pathogenicity of CNVs leading to CHD, we constructed coexpression network for CHD associated CNV-lncRNAs and protein coding genes using human organ developmental data. Our results suggested that known CHD genes might be regulated by multiple lncRNAs within both non-syndromic and syndromic CHD associated CNV regions. For the non-syndromic black module that mostly enriched in non-syndromic CHD genes, we validated the regulatory roles of a hub CNV-lncRNA *HSALNG0104472* within the 15q11.2 region. It was revealed that *HSALNG0104472* should be a main effector responsible for

cardiac defects of 15q11.2 deletion through regulating cardiomyocyte differentiation. Our results highlighted the potential contribution of lncRNAs to the pathogenicity of CHD associated CNVs.

## Methods

**Study design**. We aimed to reveal the potential contribution of lncRNAs to the pathogenicity of CHD associated CNVs. Non-syndromic and syndromic CHD associated CNVs were retrieved from CHDGKB[39] and a recent review over CHD[4], respectively. We firstly restricted our analysis on the recurrent non-syndromic CNVs which were reported in at least three cases. We constructed coexpression networks for the 568 non-syndromic CNV-lncRNAs and 19,957 protein coding genes using the human organ developmental transcriptoomic data ($n = 313$) from LncExpDB[23]. Gene expression matrixes (tpm value) of the all 313 samples were used for WGCNA, which enabled the identification of robust coexpression relationships involving CNV-lncRNAs.

To reveal the molecular basis underlying the differences between non-syndromic and syndromic CHD, we performed another WGCNA with 1500 non-syndromic and syndromic CHD associated CNV-lncRNAs. We also identified differentially expressed CHD genes and CNV-lncRNAs between the brain ($n = 87$) and heart ($n = 50$) samples.

The identified WGCNA modules were correlated to sex, developmental stage, and the seven organs with WGCNA R package. We mainly focused on the modules that were significantly correlated with heart tissues. *HSALNG0104472*, a hub CNV-lncRNA within 15q11.2, showed high coexpression and correlation with multiple key CHD genes. We then performed a series of cell experiments to validate its potential regulatory effects in cardiomyocytes.

**Ethics statement**. The iPSC line was generated from a healthy woman in our previous study[24]. Informed consent was obtained from the participant. The Ethics Committee of the SCMC reviewed and approved this study (SCMCIRB-K2022182-1). All procedure performed in studies involving human participants were in accordance with the ethical standards of the institutional and/or national research committee and with the 1964 Declaration of Helsinki and its later amendments r comparable ethical standards.

**CNV-lncRNA retrieval and weighted gene coexpression network analysis**. Non-syndromic CNVs were collected from CHDGKB[39]. Since recurrent CNVs had much more clinical implications, we only used the 19 CNVs that reported in at least three cases for our analysis (Table 1). Additionally, 21 syndromic CNVs were summarized from the recent review[4] for comparative analysis. The annotated lncRNAs[23,31] that mapped to these genomic regions with a maximum expression value (transcripts per million, tpm) ≥ 1 in the 50 heart developmental samples were considered as expressed during heart development. BEDTools v2.29.2 was used for CNV-lncRNA retrieval[40]. Processed gene expression data involving these genes for the 313 human organ developmental tissue samples provided by LncExpDB was used for coexpression analysis. The R package WGCNA v1.70[22] was used for coexpression network construction and module identification. Pearson correlation was used to calculate pairwise gene expression correlations and module-trait correlations. For module-trait correlations, adjusted $P$ value was calculated with corPvalueStudent function in WGCNA R package. The power value was set at six for coexpression network construction. The parameters for the blockwiseModules function were as follows: maxBlockSize = 6000, TOMType = "unsigned", minModuleSize = 30, reassignThreshold = 0, mergeCutHeight = 0.25, numericLabels = T and pamRespectsDendro = F. The module membership (MM) value, which was used as an estimate of the correlation between a gene/lncRNA and the corresponding module eigengene, was used to define hub CNV-lncRNAs. Protein coding genes in each module were used as the input for Gene ontology (GO) enrichment analysis and Kyoto Encyclopedia of Genes and Genomes (KEGG) pathway enrichment analyses with the R package clusterProfiler v3.10[41].

**lncRNA-miRNA-mRNA regulatory network analysis**. We retrieved 598537 lncRNA-miRNA and 331604 miRNA-mRNA interaction evidences from LncBook[25], NPInter[42], miRTarBase[43], and TarBase[44]. Integrating the interactions evidences with gene expression data of the heart developmental samples ($n = 50$), we constructed the lncRNA-miRNA-mRNA regulatory network using LncmiRSRN v3.0[34] to estimate the contribution of ceRNA mechanism to the pathogenicity of non-syndromic CHD associated CNVs.

**In vitro cardiomyocyte differentiation with iPSC**. The iPSC line generated from a healthy woman in our lab[24] was used for cardiomyocyte differentiation experiments. The iPSCs ($1 \times 10^6$) were inoculated in 6-well plates precoated with Matrigel (BD Bioscience, Heidelberg, Germany). The iPSCs were seeded and cultured with E8 medium. When the iPSCs reached the confluence of 80%~90%, in vitro induced cardiomyocyte differentiation was performed with the CardioEasy kit (Cellapybio, Beijing, China) following the protocol. The cardiomyocyte differentiation efficiencies were quantified with flow cytometry (See section "Evaluation of the efficiency of cardiomyocyte differentiation").

**Immunofluorescent staining**. For immunofluorescent staining, cells were fixed with PBS containing 4% paraformaldehyde for 20 minutes at room temperature. After washing with PBS, cells were blocked for 30 minutes with PBS containing 5% bovine serum. Staining with primary antibodies: cardiac troponin I (cTnI) (RRID: AB_2532494, ThermoFisher, Waltham, MA, USA), α-Actinin (RRID: AB_2692251, ThermoFisher, Waltham, MA, USA) diluted in blocking buffer was performed for overnight at −4 °C temperature. Secondary antibodies were used in the next day, following staining with DAPI to detect the cell nucleus. Fluorescent images were acquired using a Laser confocal microscope (Leica TCS SP8).

**RNAseq analysis**. Total cellular RNA was isolated with TRIzol reagent (Ambion, Austin, TX, United States) according to the protocol. RNA concentration and integrity were measured using the Agilent Bioanalyzer 2100 system. Samples with RIN ≥ 7 were qualified for sequencing library construction. We prepared the sequencing libraries using the mRNA-seq Lib Prep Kit for Illumina (ABclonal Technology Co., Wuhan, China) with adapters for the BGI platform. Sequencing was performed on the DNBSEQ-T7. The data were processed as follows: quality analysis and base quality filtering with FastQC v0.11.9 (https://www.bioinformatics.babraham.ac.uk/projects/fastqc) and Trim Galore v0.6.6 (https://github.com/FelixKrueger/TrimGalore), rRNA removing with SortMeRNA v4.3.4[45], alignment with STAR v2.7.10[46], reads counting with featureCounts[47] implemented in the Subread package v2.0.3, differential expression analysis and principal component analysis with DESeq2 v1.32[48], and functional enrichment analysis with clusterProfiler v3.10[41]. For differential expression analysis, gene raw counts were normalized with rlog function in DESeq2 v1.32[48].

**Human organ developmental data**. The gene expression data from LncExpDB[23] were generated through comprehensive annotation for lncRNAs of the original dataset of developmental samples ($n = 313$, brain: 55, cerebellum: 59, heart: 50, kidney: 40, liver: 50, ovary: 18, and testis: 41) collected from ArrayExpress (E-MTAB-6814). The tissue sampling started at four weeks post-conception and then sampled each week until 20-week post-conception except for 14-, 15-, and 17-week post-conception. Postnatal tissues were sampled as neonates, infants (6–9 months), toddlers (2–4 years), school age (7–9 years), teenagers (13–19 years), and adults (~65 years). The kidney development was sampled until 8 years of age and the ovary development was only sampled prenatally[49].

**In vitro cardiomyocyte differentiation data**. The processed transcriptomic data ($n = 297$) of in vitro cardiomyocyte differentiation from LncExpDB[23] was used to validate the highly correlated coexpression relationship between the CNV-lncRNAs and CHD genes in the non-syndromic black module identified by WGCNA. The original data was collected from the Gene Expression Omnibus (GEO) under accession GSE122380. The original data was generated from cardiomyocyte differentiation of iPSC lines from 19 individuals from the Yoruba HapMap population[50].

**Quantification of sample traits**. Sample traits in human organ developmental data were collected from LncExpDB[23]. Categorical variables (Sex and Tissue) of samples were quantized into 1 and 0. For sex, male was quantized into 1 and female was quantized into 0. For 7 organs (Heart, for example), heart samples were quantized into 1 while other samples were quantized into 0. Continuous variables (Developmental stage) of samples were quantized into week according to the following rules: For embryo samples (before birth), the developmental stage value equal how many weeks they were post conception (For example, if the embryo was 10 weeks old, the value was 10.). For samples collected after birth, the developmental stage value equal 40 (human pregnancy estimated value of 40 weeks) plus age (count in weeks). 1 year was calculated as 52 weeks, 1 month as 1/12 years, and 1 week as 7 days. (For example, for a sample at 6 months after birth, the development stage value was $40 + 6/12 \times 52 = 66$; for a sample at 7 days after birth, the development stage value was $40 + 7/7 = 41$.) (Supplementary Data 1).

**Identification of association between CNV-lncRNA and CHD subtype**. Details (including CHD subtype) of recurrent non-syndromic CHD associated CNVs were collected from CHDGKB[39]. After filtering the CNV records without specific CNV coordinate information, the original coordinates of each CNV record were converted to the hg38 assembly using liftover tool in UCSC (http://genome.ucsc.edu)[51]. Coordinates of all CNV-lncRNAs (hg38 assembly) were retrieved from LncExpDB[23]. We took the intersection of coordinates to identify the association between CNV-lncRNA and CHD subtype.

**Relative weight analysis**. Relative weight analysis (RWA), which was also known as driver analysis, was performed with RWA Web and R package[52] to identify key driver CNV-lncRNAs in the non-syndromic black module. For each RWA analysis, the CHD gene was used as response variable and CNV-lncRNAs were used as predictors. $R^2$ estimated the contribution of a group of predictors to the response variable. When 0 was not included in the confidence intervals, the relative weight was denoted significant. Otherwise, the relative weight was not significantly from one another. See more details in the RWA Web.

**HSALNG0104472 overexpression and knockdown in cardiomyocyte cell lines**. For the overexpression experiments, full length of the transcript (HSALNT0217290) for HSALNG0104472 was cloned into the lentiviral vector PCDH-CMV-MCS-EF1a-gfp-T2A-puro, the empty vector was used as control. For the stable knockdown experiments using shRNA, a complementary sequence (AGGGAACCAGCTTCAGAACTCAAGAGGTTCTGAAGCTGGTTCCCTTT TTTT) targeting the HSALNG0104472 transcript was inserted into the lentiviral vector pLVX-shRNA2-zsGreen-PGK-puro. The corresponding scramble sequence (CGTATACCCGGAACAAAGGTCAAGAGCCTTTGTTCCGGGTATACGTT TTTT) was used as the control. The constructed lentiviral plasmids were respectively transfected into HEK293 cells with the packaging plasmids psPAX2 and pMD2·G by Lipofectamine 2000 (Invitrogen, Carlsbad, CA, USA) to produce virus. AC16/iPSCs were infected with the lentiviruses following puromycin treatment for several days to obtain the overexpression and knockdown cell lines. Knockdown experiments were also repeated through transient transfection with a Smart Silencer (mixture of small interfering RNAs and antisense oligonucleotides targeting HSALNG010447, synthesized by RiboBio, Guang Zhou, China). The Smart Silencer was transfected in iPSCs or cardiomyocytes with the Lipofectamine RNAiMAX (Invitrogen, Carlsbad, CA, USA) The silencer sequences were listed in Supplementary Table 3.

**Quantitative reverse transcription qPCR analysis**. A total of 1 µg cellular RNA was used as the template for cDNA preparation with the PrimeScript RT Reagent Kit (Takara, Dalian, China). Quantitative reverse transcription qPCR (RT-qPCR) was performed with the TB Green Premix Ex Taq II kit (Takara, Dalian, China) on the CFX 96 Real-Time PCR detection system (Bio-Rad Laboratories, Inc., Hercules, CA, United States). Relative gene expression levels were calculated based on the $2^{-\Delta\Delta Ct}$ method. At least three biologically independent experiments were conducted for each group. GAPDH was used as the internal reference gene. The RT-qPCR primer pairs were listed in Supplementary Table 4.

**Evaluation of the efficiency of cardiomyocyte differentiation**. Flow cytometry was used to evaluate the relative yield of cardiomyocytes and efficiency of cell differentiation by measuring the number of cells expressing cardiac-specific proteins cardiac Troponin T (cTnT). Cardiac Troponin T Polyclonal Antibody PE Conjugated (Catalog No: #C90559PE) was used to target cardiomyocytes. For sample preparation, digest cardiomyocytes with mild digestive enzymes for 5 min. Add an equal volume of cell culture medium to stop the reaction. Centrifuge at 25 °C 800 × g for 5 mins and resuspend cells with PBS to wash the cells 2 times at the same procedure. Aliquot $1 \times 10^6$ cells from pretreated sample in 100 ul by volume in a 5 ml assay tube per test. Add Cardiac Troponin T Polyclonal Antibody PE Conjugated at the appropriate dilution (1:100) to the assay tubes. Incubate at 37 °C for 1 hour. Wash by centrifugation in 2-3 ml wash buffer (PBS with 0.1% FBS). Resuspend cells in 0.3–0.5 ml PBS and analyze on flow cytometer. BD FACSCanto™ flow cytometer was used for cardiomyocyte differentiation assays. The percentage of cells was assessed by BD FACSDiva™ v8.0.1 Software. Abundance of positive and negative fractions were determined by cell counting post sort. Gating strategy was shown in Supplementary Information (Supplementary Fig. 10).

**Statistics and reproducibility**. A threshold of adjusted P value < 0.05 was used for differential gene expression analysis. The P values were adjusted for multiple testing using the Benjamini-Hochberg method. Hypergeometric tests were used to estimate the significance of enrichment of known CHD genes in coexpression modules. Two-tailed Student's t test was used for comparison between two groups. At least three biologically independent experiments were conducted for each group.

**Reporting summary**. Further information on research design is available in the Nature Portfolio Reporting Summary linked to this article.

## Data availability

The RNAseq raw data for overexpression and knockdown experiments on HSALNG0104472 in AC16 cardiomyocytes that support the findings of this study are openly available in GEO: GSE201076. All data are available in the main text or the supplementary materials. The plasmid for HSALNG0104472 overexpression and knockdown has been deposited in Addgene (197902, 197989, and 197991).

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

## Acknowledgements

We thank all the participants for their cooperation in this research. We also thank Ms MJ Zhou at Fresenius Medical Care for editing the English text of a draft of this manuscript. This work was supported by National Key R&D Program of China [2021YFC2701104 to Qh.F.]; National Natural Science Foundation of China [82172352 to B.W., 82072372 to Qh.F.]; Pudong Science Technology and Economy Commission [PKJ2022-Y04 to B.W.]; National Facility for Translational Medicine (Shanghai) [TMSK-2021-133 to B.W.]; and Shanghai Municipal Science and Technology [20JC1418500, 21Y31900300 and 20dz2260900 to Qh.F.].

## Author contributions

Yb.L. and B.W. contributed literature review, study design, data collection, data analysis, data interpretation, and drafting the manuscript; Q.F., M.Q., X.L., Yw.L., and X.Z. contributed data collection, experimental validation, data interpretation, and manuscript preparation; Y.X., Qh.F., and B.W. contributed supervision of all aspects of the study and manuscript preparation. The authors read and approved the final manuscript.

## Competing interests

The authors declare no competing interests.
