## [Peer Review File · Communications Biology]

Reviewers' comments:

Reviewer #1 (Remarks to the Author):

This is an original research article by Lu, Y et al. Athores investigated the potential role of CNV-lncRNAs in congenital heart defects (CHDs). They extracted the CNV-lncRNA from nonsyndromic CHDs and utilized WGCNA method to construct integrated coexpression profile of such CNV-lncRNAs and protein coding genes of human organ developmental transcriptomic data from LncExpDB. From these analyses, three modules significantly correlated with heart tissues, one of which showed enrichment of known CHD genes were identified, also enriched with CNV-lncRNAs. Among several hub CNV-lncRNAs, the focused on HSALNG0104472, the hub CNV-lncRNA in the black module that is located in the 15q11-2 CNV region. This region proved to be associated with total anomalous pulmonary venous connection (TAPVC) in a previous work by the authors. By conducting gain and loss of function studies in fetal rat and adult cardiomyocytes, they validated the potential regulatory effect of HSALNG0104472 on the known CHD genes (GATA6, NKX2:5), suggesting the potential causal role of HSALNG0104472 in CHD. Following similar approaches, they performed comparative analyses of syndromic and non-syndromic CHD associated CNV-lncRNAs, suggesting shared genetic basis between CHD and neurodevelopmental disorders.

Overall, the article addresses an important question (lncRNA role in CHDs) which may improve our understanding of significant portion of CHD. The authors utilized appropriate bioinformatic tools and systems genetic approaches to answer this question. The data is well-presented and the scientific rigor is good. However, I have 2 major concerns:

1. Authors used low Pearson correlation coefficient cut off [r] value of [0.2] to define significance of heart modules, even with significant P values. Higher [r] values should be used to define appropriate module-organ (heart) correlation.
2. The mechanistic validation is restricted to in vitro cell model. Therefore, the data remain suggestive/speculative. Establishing the causal role of HSALNG0104472 in TAPVR/or other CHD phenotype requires: a) in vivo validation. b) establishing interaction with GATA4 or NKX 2:5; c) ruling out the impact of other protein coding genes located within the same CNV region, etc.

Minor Comments:

1. Fig 7c. The article will benefit from quantifying Western analysis.
2. Authors mentioned lncRNA classes in their discussion. However, they did not provide classification of the CNV-lncRNA dataset. Suggest to add.

Reviewer #2 (Remarks to the Author):

This study investigated whether lncRNAs within CNVs could contribute to the etiology of CHD associated CNVs. Based on co-expression network analysis of human heart developmental transcriptomic data and experimental validation, the authors revealed that lncRNA HSALNG0104472 could regulate the expression of CHD genes such as NKX2-5 and GATA6 in fetal cardiomyocytes. As only part of disease associated CNVs could be interpreted as dosage effect due to disruption of coding genes, the shift to lncRNAs would provide new insights into the elucidation of CNV associated etiologies in a wide range of diseases.

However, my major concern is that this study validated the regulation role in rat fetal cardiomyocyte cell line (H9C2) by overexpression a human lncRNA. If the rat genome doesn't express any HSALNG0104472 homolog, the regulation role could hardly be convincing.

Another major concern is that CHD is frequently occurred as syndromic, why this study focused on non-syndromic CHD associated CNVs?

The authors emphasize the correlation between CHD and neurodevelopmental disorders, while according to Figure 6, the correlation between heart and liver is also very obvious.

In addition, there are some minor concerns as follows.

1. As 4 CNVs are associated with both syndromic and non-syndromic CHD, 36 rather than 33 CNVs should be listed in figure 2a;
2. It seems that figures 3a and 3c represent similar results;
3. It is not clear how to connect syndromic yellow module with the non-syndromic black module.

Reviewer #3 (Remarks to the Author):

In this study, the authors aimed to investigate whether long noncoding RNAs (lncRNAs) within CNVs (CNV-lncRNAs) could contribute to the etiology of CHD associated CNVs. Using the human organ developmental transcriptomic data, they constructed coexpression networks involving CNV-lncRNAs within CHD associated CNVs and protein coding genes.

Although the data analysis shows some interesting data, but the validation experiments are insufficient.

Major

1. It is not possible to detect developmental abnormalities in cell line experiments; the creation of KO mice of HSALNG0104472 would provide a clue for analysis.
2. In Figure 7, Nkx2.5 and GATA6 mRNAs are altered by lncRNA overexpression, but not at the protein level. How do these molecules affect downstream signaling?
3. Cardiomyocytes are not migratory cells. Why is a migration assay being performed in Figure 7f?

Reviewer #4 (Remarks to the Author):

The authors present a manuscript exploring the relationship between lncRNAs and CNVs that may be associated with CHD. Given the complex and unclear etiology of a large proportion of CHD, this topic is certainly of interest – the authors present this well in the introduction, summarizing recent literature and identifying that the genetic basis of most CHD remains undetermined. Although the analyses presented in the manuscript are complex, the “big picture” conclusions and major points should be clarified and emphasized.

1. It is unclear in the manuscript why lncRNAs located in CNV regions (CNV-lncRNAs) were specifically evaluated, as opposed to all lncRNAs globally? As mentioned in the discussion (p.13, line 388), lncRNAs may repress or activate gene expression in cis or trans. Would it be possible for lncRNAs that aren't specifically located in a CNV region to cause CHD pathogenesis? Clarification regarding the rationale for specifically evaluating CNV-lncRNAs would be helpful.
2. The separation of syndromic and non-syndromic CHD is a strength. Nevertheless, as illustrated by Table 1, there is a wide variety in the anatomic defects and severity included under the umbrella of “CHD”. For example, the embryological defect in TOF is distinct from TAPVC or HRH. Would additional conclusions be able to be made if a single CHD type was considered, instead of CHD as a whole?
3. In the methods, p. 15 line 474, the manuscript states that an “empty vector was used as a control” but a scramble sequence is also listed. Is there data from the scramble sequence transfection? Is

there a reason that the scramble sequence is not used as a control?

4. Given the species difference between H9C2 and AC16 cell lines, it is unclear whether the differences seen with overexpression and knockdown of HSALNG0104472 are simply related to developmental stage/maturity. Was there consideration of using a human induced pluripotent stem cell derived cardiomyocyte line?

5. It is confusing to use the ASD abbreviation for autism spectrum disorder since ASD is a common abbreviation for atrial septal defect. Consider writing out autism spectrum disorder instead of using the abbreviation to avoid confusion.

Responses point-to-point are as below:

Reviewer #1 (Remarks to the Author):

This is an original research article by Lu, Y et al. Authors investigated the potential role of CNV-lncRNAs in congenital heart defects (CHDs). They extracted the CNV-lncRNA from nonsyndromic CHDs and utilized WGCNA method to construct integrated coexpression profile of such CNV-lncRNAs and protein coding genes of human organ developmental transcriptomic data from LncExpDB. From these analyses, three modules significantly correlated with heart tissues, one of which showed enrichment of known CHD genes were identified, also enriched with CNV-lncRNAs. Among several hub CNV-lncRNAs, the focused on HSALNG0104472, the hub CNV-lncRNA in the black module that located in the 15q11.2 CNV region. This region proved to be associated with total anomalous pulmonary venous connection (TAPVC) in a previous work by the authors. By conducting gain and loss of function studies in fetal rat and adult cardiomyocytes, they validated the potential regulatory effect of HSALNG0104472 on the known CHD genes (GATA6, NKX2:5), suggesting the potential causal role of HSALNG0104472 in CHD. Following similar approaches, they performed comparative analyses of syndromic and non-syndromic CHD associated CNV-lncRNAs, suggesting shared genetic basis between CHD and neurodevelopmental disorders.

Overall, the article addresses an important question (lncRNA role in CHDs) which may improve our understanding of significant portion of CHD. The authors utilized appropriate bioinformatic tools and systems genetic approaches to answer this question. The data is well-presented and the scientific rigor is good. However, I have 2 major concerns:

1. Authors used low person correlation coefficient cut off [r]value of [0.2] to define significance of heart modules, even with significant P values. Higher [r] values should be used to define appropriate module-organ (heart)correlation.

R: We appreciate for your valuable comments. We agree with this suggestion and have set the threshold value of the coefficients as 0.6 for module-organ correlation. Descriptions has been modified throughout the text. The red dashes that indicate threshold value of the coefficients in Figure 2b) have been modified to 0.6. Description of modules with correlation of coefficients have been removed (line 104, Figure 2c).

[Changes in Figure 2: The red dashes that indicate threshold value of the coefficients in Figure 2b) have been modified to 0.6.]

[Changes in Figure 6: The coefficients shown in Figure 6 have been modified accordingly.]

2. The mechanistic validation is restricted to in vitro cell model. Therefore, the data remain suggestive/speculative. Establishing the causal role of HSALNG0104472 in TAPVR/or other CHD phenotype requires: a) in vivo validation. b) establishing interaction with GATA4 or NKX 2:5; c) ruling out the impact of other protein coding genes located within the same CNV region, etc.

R: We thank the reviewer for this comment. We have performed additional experimental validation for *HSALNG0104472*:

a) Since *HSALNG0104472* is not conserved in model animals, we have conducted the iPSCs-cardiomyocyte differentiation experiments to simulate in vivo validation for cardiac development. Our results showed that knockdown of *HSALNG0104472* would significantly affect cardiomyocyte differentiation (Figure 8; Supplementary Movie 1, 2). In Figure 8, we have added the results of additional validation of *HSALNG0104472*-knockdown effect through iPSCs-cardiomyocyte differentiation.

[Changes in Figure 8: Additional validation of *HSALNG0104472*-knockdown effect through iPSCs-cardiomyocyte differentiation.]

b) We really appreciated for this suggestion. Undoubtedly, specific mechanism between CNV-lncRNAs and target protein coding genes was worthy of study. However, we aimed to validate the potential regulatory effects of CNV-lncRNAs to cardiac development through affecting well-documented CHD genes in this study. Such regulatory relationships were not necessarily established through direct interaction between lncRNA and these coexpressed protein coding genes, but maybe with the participation of more intermediate molecules in the transcriptional regulation. Therefore, we did not focus on specific interaction mechanism in present study.

c) We thank the reviewer for providing this comprehensive viewpoint. *HSALNG0104472* locates in 15q11.2 of human genome, CNVs of which mainly encompass 4 protein coding genes. We have added additional discussion of these genes in the text (the 6th paragraph of discussion, line 344-352).

“The 15q11.2 deletion region mainly encompasses 4 protein coding genes: *TUBGCP5*, *CYFIP1*, *NIPAI*, and *NIPA2*. In this study, our differential gene expression analysis between developing heart and brain tissues indicated that *NIPAI* is preferentially expressed in developing brain tissues, whereas the other three genes did not show significant differential expression. Since only *TUBGCP5* and *NIPAI* were reported to be expressed in fetal hearts. We previously created *TUBGCP5* knockout iPSCs and proved that reduction of *TUBGCP5* would affect cardiomyocyte differentiation²⁴. Here, we proved that the CNV-lncRNA *HSALNG0104472* specifically express in developing heart tissues and its reduction would generate more severe impact on cardiomyocyte differentiation. Therefore, *HSALNG0104472* should be a major effector for cardiac defects resulted from 15q11.2 deletion.”

Minor Comments:

1. Fig 7c. The article will benefit from quantifying Western analysis.

R: We thank the reviewer for pointing out this issue. Previously, we did three replicates for quantifying the protein expression but found no significant change.

Whereas, because other reviewers suggested that overexpression of the lncRNA *HSALNG0104472* in H9C2 cell line should not be convincing since rat does not express any *HSALNG0104472* homolog, we decided to remove these results in updated manuscript.

2. Authors mentioned lncRNA classes in their discussion. However, they did not provide classification the CNV-lncRNA dataset. Suggest to add.

R: We thank the reviewer for this suggestion. In the fifth paragraph of the discussion part, we mentioned that the lncRNA could be categorized as signaling lncRNAs, decoy lncRNAs, guide lncRNAs, scaffold lncRNAs, and enhancer lncRNAs. Such mechanical classification was defined by experimental evidences. However, most of the CNV-lncRNAs we identified have not been investigated experimentally, thus we added the classification information reflecting the CNV-lncRNAs' locations related to protein coding genes (i.e. sense, antisense, intronic, intergenic). Additionally, we also added the statistics of sequence conservation for these CNV-lncRNAs (Figure 2d, 2e); Supplementary Figure 5, 6; Supplementary Data 1, 3).

[Changes in Figure 2: Threshold line was changed to 0.6 in Figure 2b; description of modules with low correlations coefficients was removed from Figure 2c; classification and conservation information for the CNV-IncRNAs of the black module was added as Figure 2 d, e.]

[Changes in Supplementary Figure 5: Classification information of CNV-lncRNAs was added.]

[Changes in Supplementary Figure 6: Sequence conservation information of CNV-lncRNAs was added.]

Reviewer #2 (Remarks to the Author):

This study investigated whether lncRNAs within CNVs could contribute to the etiology of CHD associated CNVs. Based on co-expression network analysis of human heart developmental transcriptomic data and experimental validation, the authors revealed that lncRNA HSALNG0104472 could regulate the expression of CHD genes such as NKX2-5 and GATA6 in fetal cardiomyocytes. As only part of disease associated CNVs could be interpreted as dosage effect due to disruption of coding genes, the shift to lncRNAs would provide new insights into the elucidation of CNV associated etiologies in a wide range of diseases.

However, my major concern is that this study validated the regulation role in rat fetal cardiomyocyte cell line (H9C2) by overexpression a human lncRNA. If the rat genome doesn't express any HSALNG0104472 homolog, the regulation role could hardly be convincing.

R: We thank the reviewer for pointing out this issue, and we have realized that it should not be appropriate to assess the function of a specific lncRNA in a distant related species that lack its homolog. We have removed the results assayed in the H9C2 cell line. Instead, we have conducted the human iPSCs-cardiomyocyte differentiation experiments to simulate in vivo validation for cardiac development. Our results showed that knockdown of *HSALNG0104472* would significantly affect cardiomyocyte differentiation (Figure 8; Supplementary Movie 1, 2).

[Changes in Figure 8: Additional validation of *HSNNG0104472*-knockdown effect through iPSCs-cardiomyocyte differentiation.]

Another major concern is that CHD is frequently occurred as syndromic, why this study focused on non-syndromic CHD associated CNVs?

R: We thank the reviewer's comment. Since CHD is the most common birth defect and CNVs have been identified as important sources of causal factors for CHD, we aimed to identify the lncRNAs that potentially affect cardiac development due to CNV. Although CHD is frequently occurred as syndromic, there should be still quite proportion of CHD cases should be non-syndromic and could not be explained by existing genetic evidences. We believe that the separate analysis of non-syndromic and syndromic CHD associated CNVs would better elucidate the phenotypic

correlations for these CNV-lncRNAs.

The authors emphasize the correlation between CHD and neurodevelopmental disorders, while according to Figure 6, the correlation between heart and liver is also very obviously.

R: We thank the reviewer for pointing out this issue. Other reviewers suggested that higher coefficient [r] threshold should be used to define appropriate module-organ correlation. We agree with this suggestion and have set the threshold value of the coefficients as 0.6 to define module-organ correlation. Descriptions has been modified throughout the text. The coefficients shown in Figure 6 have been modified accordingly. Although significant module-organ correlation indicated that genes in the module specifically expressed in certain tissue, pathway enrichment analyses results (Figure 4 e); Supplementary Data 2) and hypergeometric distribution test results of CHD and autism spectrum disorder gene sets (Figure 4d; Supplementary Data 9) show that the turquoise module, in which genes expressed with no obvious tissue specificity, was more likely to affect multi-system development, which was worthy of follow-up study.

[Changes in Figure 4: Figure 4e was added to indicate enrichment of cardiovascular development related pathways in the black and turquoise modules.]

[Changes in Figure 6: The coefficients shown in Figure 6 have been modified accordingly.]

In addition, there are some minor concerns as follows.

1. As 4 CNVs are associated with both syndromic and non-syndromic CHD, 36 rather than 33 CNVs should be listed in figure 2a;

R: We apologize for the confusion generated by combining continuous CNVs for labeling (1q41-q42, 1q43-q44, 20p12-p13). We have re-labeled the 36 CNVs in Figure 2a.

[Changes in Figure 2: The 36 CNVs were relabeled in Figure 2a.]

2. It seems that figures 3a and 3c represent similar results;

R: We apologize for the confusion. Previously we only illustrated in Figure 3 caption that correlation in Figure 3a (shown in lines) were calculated based on human organ developmental dataset (n=313), while correlation in Figure 3c (shown in dots) were calculated based on In vitro cardiomyocyte differentiation dataset (n=297). We have moved Figure 3c to the supplemental materials in the updated manuscript.

[Changes in Figure 3: Figure 3c has been moved to Supplementary Figure 7.]

3. It is not clear how to connect syndromic yellow module with the non-syndromic black module.

R: We thank the reviewer's comment. We used 19957 protein coding genes in the WGCNA for both syndromic and non-syndromic CHD associated CNVs. 84.89% (579/682) of the genes clustered in the non-syndromic black module appear in the syndromic yellow module. The two modules showed highest correlations with heart tissues and were enriched in the same functions. We have added additional description for their connection (the 8th paragraph of results, line 219-223).

“The syndromic yellow module (s-yellow) should be corresponding to the non-syndromic black module: 1) the s-yellow module showed the highest correlations with heart tissue ($r_{\text{heart}}=0.83$) (Figure 2, 6); 2) most (84.89%, 579/682) of the genes in non-syndromic black module appeared in the s-yellow module; 3) Genes in the two modules were enriched in the same functions (Supplementary Data 8).”

Reviewer #3 (Remarks to the Author):

In this study, the authors aimed to investigate whether long noncoding RNAs (lncRNAs) within CNVs (CNV-lncRNAs) could contribute to the etiology of CHD associated CNVs. Using the human organ developmental transcriptomic data, they constructed coexpression networks involving CNV-lncRNAs within CHD associated CNVs and protein coding genes.

Although the data analysis shows some interesting data, but the validation experiments are insufficient.

Major

1. It is not possible to detect developmental abnormalities in cell line experiments; the creation of KO mice of *HSALNG0104472* would provide a clue for analysis.

R: We thank the reviewer's comment. It is a great pity that the mice do not have a homolog of *HSALNG0104472* according to the sequence conservation analysis (Supplementary Data 3). Instead, we have performed additional cardiomyocyte differentiation experiments to simulate cardiac developmental and beating behavior abnormalities after *HSALNG0104472* reduction using human iPS cells (Figure 8).

[Changes in Figure 8: Additional validation of *HSALNG0104472*-knockdown effect through iPSCs-cardiomyocyte differentiation.]

2. In Figure 7, Nkx2.5 and GATA6 mRNAs are altered by lncRNA overexpression, but not at the protein level. How do these molecules affect downstream signaling?

R: We thank the reviewer's comment. Since another reviewer raised the issue that it should not be appropriate to assess the function of a specific lncRNA in a distant related species that lack its homolog, we have removed the results generated by H9C2 cell line experiments in Figure 7. Instead, we performed *HSALNG0104472*-knockdown experiments using iPSCs and assessed its effect on cardiomyocyte differentiation. The results have been added as mentioned in previous response (Figure 8).

3. Cardiomyocytes are not migratory cells. Why is a migration assay being performed in Figure 7f?

R: We thank the reviewer for pointing out this issue. We have realized that it should not be appropriate to evaluate cell migratory using a cardiomyocyte cell line. We have removed these results in the updated manuscript.

Reviewer #4 (Remarks to the Author):

The authors present a manuscript exploring the relationship between lncRNAs and CNVs that may be associated with CHD. Given the complex and unclear etiology of a large proportion of CHD, this topic is certainly of interest – the authors present this well in the introduction, summarizing recent literature and identifying that the genetic basis of most CHD remains undetermined. Although the analyses presented in the manuscript are complex, the “big picture” conclusions and major points should be clarified and emphasized.

1. It is unclear in the manuscript why lncRNAs located in CNV regions (CNV-lncRNAs) were specifically evaluated, as opposed to all lncRNAs globally? As mentioned in the discussion (p.13, line 388), lncRNAs may repress or activate gene expression in cis or trans. Would it be possible for lncRNAs that aren't specifically located in a CNV region to cause CHD pathogenesis? Clarification regarding the rationale for specifically evaluating CNV-lncRNAs would be helpful.

R: We thank the reviewer's suggestion. We have added additional clarification regarding the rationale for specifically evaluating CNV-lncRNAs (paragraph 2 of the introduction section, line 52-55).

“CHD is characterized by its high genetic heterogeneity, which made the discovery of disease causing lncRNAs frustratingly difficult. Yet, benefited from the accumulated evidences that established the robust association of CNVs with CHD, recurrent pathogenic CNVs have provided a natural source to link lncRNAs to disease phenotypes.”

2. The separation of syndromic and non-syndromic CHD is a strength. Nevertheless, as illustrated by Table 1, there is a wide variety in the anatomic defects and severity included under the umbrella of “CHD”. For example, the embryological defect in TOF is distinct from TAPVC or HRH. Would additional conclusions be able to be made if a single CHD type was considered, instead of CHD as a whole?

R: We appreciated for the reviewer's suggestion. We compiled the genomic regions that have been identified as CHD associated CNVs for global screening of CNV-lncRNAs that contribute to congenital cardiac defects. CHD is characterized as

highly genetic heterogeneous with variable expressivity. Whereas, we agree that specifically reveal the relationship of genetic-phenotype would help understanding the underlying molecular pathogenesis of CHD. Conclusions regarding specific CHD types require high quality clinical and genetic evidences. Since many reported cases lack precise genomic ranges for the identified CNVs (Supplementary Data 1), we have tried to add the potential phenotypic relationship for several CNV-lncRNAs with explicit CNV genomic regions (line 153, Figure 3b).

[Changes in Figure 3: The evident relationships between CNV-lncRNA and CHD subtypes was added in Figure 3b.]

3. In the methods, p. 15 line 474, the manuscript states that an “empty vector was used as a control” but a scramble sequence is also listed. Is there data from the scramble sequence transfection? Is there a reason that the scramble sequence is not used as a control?

R: We apologize for the confusion generated by the misleading description in the method part. We used empty vector as a control for the overexpression experiment, which was performed in the rat fetal cardiomyocyte cell line. For the knockdown experiment, shRNA was used to target *HSALNG0104472* and the scramble sequence was used as control. Since the rat cell lines does not express any homolog for *HSALNG0104472*, we have removed the results derived from H9C2 cell line in the updated manuscript. We have also rephrased the methods.

4. Given the species difference between H9C2 and AC16 cell lines, it is unclear whether the differences seen with overexpression and knockdown of *HSALNG0104472* are simply related to developmental stage/maturity. Was there consideration of using a human induced pluripotent stem cell derived cardiomyocyte line?

R: We thank the reviewer for pointing out this issue. We have realized that it should not be appropriate to assess the function of *HSALNG0104472* in a rat cell line. We have performed knockdown experiments of *HSALNG0104472* in iPS cells to evaluate its effects on cardiomyocyte differentiation (Figure 8; Supplementary Movie 1,2).

[Changes in Figure 8: Additional validation of *HSALNG0104472*-knockdown effect through iPSCs-cardiomyocyte differentiation.]

5. It is confusing to use the ASD abbreviation for autism spectrum disorder since ASD is a common abbreviation for atrial septal defect. Consider writing out autism spectrum disorder instead of using the abbreviation to avoid confusion.

R: We appreciate for pointing out this issue. We have replaced all these misleading “ASD” with autism spectrum disorder throughout the text.

Reviewers' comments:

Reviewer #1 (Remarks to the Author):

I thank the author for addressing my comments satisfactory.
No additional revisions/comments

Reviewer #2 (Remarks to the Author):

The authors responded to most of my questions in a satisfactory manner. They find the CNV-lncRNA HSALNG0104472, co-expressed with multiple key CHD genes in heart development, could influence the beating behavior of differentiated cardiomyocytes, and its reduction significantly affects cardiomyocyte differentiation. Although the molecular mechanism is not clear, these results help to understand the role of lncRNA in the etiology of CHD associated CNVs.

Reviewer #3 (Remarks to the Author):

In this study, the authors aimed to investigate whether long noncoding RNAs (lncRNAs) within CNVs (CNV-lncRNAs) could contribute to the etiology of CHD associated CNVs. Using the human organ developmental transcriptomic data, they constructed coexpression networks involving CNV-lncRNAs within CHD associated CNVs and protein coding genes. Although the data analysis shows some interesting data, but the validation experiments are still insufficient.

Major

Knockdown of HSALNG0104472 in human iPSCs has reduced expression of Nkx2-5 and TBX20, but the mechanism has not been shown at all. It is also unclear how this degree of altered expression of these transcription factors leads to abnormalities in cardiac development.

Reviewer #4 (Remarks to the Author):

Overall, the authors were responsive to the reviews. A major improvement is the removal of the rodent cell culture models and use of a human iPSC model. It is extremely challenging to see the alpha-actinin (and cTnI for the knockdown) staining in Figure 8b, which make it difficult to verify the author's conclusion that there is a lack of mature myocardial sarcomere organization in the knockdown.

Responses point-to-point are as below:

Reviewer #1 (Remarks to the Author):

I thank the author for addressing my comments satisfactory.

No additional revisions/comments

R: We thank the reviewer for the valuable comments.

Reviewer #2 (Remarks to the Author):

The authors responded to most of my questions in a satisfactory manner. They find the CNV-lncRNA *HSALNG0104472*, co-expressed with multiple key CHD genes in heart development, could influence the beating behavior of differentiated cardiomyocytes, and its reduction significantly affects cardiomyocyte differentiation. Although the molecular mechanism is not clear, these results help to understand the role of lncRNA in the etiology of CHD associated CNVs.

R: We thank the reviewer for the suggestions. We have tried to validate the regulatory effect of *HSALNG0104472* on cardiac differentiation using cell models. We totally agree that a deep investigation of the molecular mechanism of *HSALNG0104472* should help reveal understand the role of lncRNA in the etiology of CHD associated CNVs. Since the present study was originally not designed for mechanism research, we have added discussion of the unresolved molecular mechanism as limitation and future directions:

line 418-423: inconsistent regulatory effect of *HSALNG0104472* on known CHD genes was observed in iPSC and differentiated cardiomyocytes (i.e., transcriptional levels of *NKX2.5*, *MYBPC3* and *TBX20* were downregulated in iPSCs but not in cardiomyocytes), which indicated stage-specific roles of *HSALNG0104472*. It would be of interest to investigate how *HSALNG0104472* regulated the expression of key cardiac transcription factors such as *NKX2.5* and *TBX20* in iPSCs, and characterize the phenotypic effect of *HSALNG0104472* haploinsufficiency on cardiac development. Altogether, we provided evidences that CNV-lncRNAs potentially regulate expression pattern of well-established CHD genes.

Line426-428: Since most of the CNV-lncRNA are not evolutionary conserved, in vitro models such as cardiac organoids have emerged as potential tools to provide insights of molecular mechanisms for them.

Reviewer #3 (Remarks to the Author):

In this study, the authors aimed to investigate whether long noncoding RNAs (lncRNAs) within CNVs (CNV-lncRNAs) could contribute to the etiology of CHD associated CNVs. Using the human organ developmental transcriptomic data, they constructed coexpression networks involving CNV-lncRNAs within CHD associated CNVs and protein coding genes.

Although the data analysis shows some interesting data, but the validation experiments are still insufficient.

Major

Knockdown of *HSALNG0104472* in human iPSCs has reduced expression of *Nkx2-5* and *TBX20*, but the mechanism has not been shown at all. It is also unclear how this degree of altered expression of these transcription factors leads to abnormalities in cardiac development.

R: We appreciated the comments from the reviewer. A deep investigation of the molecular mechanism of *HSALNG0104472* would definitely improve our understanding of the role of lncRNAs in the etiology of CHD associated CNVs. Much greater efforts are needed to achieve such goal. Since the present study was originally not designed for mechanism research, we have discussed this part as limitations and future directions:

line 418-423: inconsistent regulatory effect of *HSALNG0104472* on known CHD genes was observed in iPSC and differentiated cardiomyocytes (i.e., transcriptional levels of *NKX2.5*, *MYBPC3* and *TBX20* were downregulated in iPSCs but not in cardiomyocytes), which indicated stage-specific roles of *HSALNG0104472*. It would be of interest to investigate how *HSALNG0104472* regulated the expression of key cardiac transcription factors such as *NKX2.5* and *TBX20* in iPSCs, and characterize the phenotypic effect of *HSALNG0104472* haploinsufficiency on cardiac development.

Altogether, we provided evidences that CNV-lncRNAs potentially regulate expression pattern of well-established CHD genes.

Line426-428: Since most of the CNV-lncRNA are not evolutionary conserved, in vitro models such as cardiac organoids have emerged as potential tools to provide insights of molecular mechanisms for them.

Reviewer #4 (Remarks to the Author):

Overall, the authors were responsive to the reviews. A major improvement is the removal of the rodent cell culture models and use of a human iPSC model. It is extremely challenging to see the alpha-actinin (and cTnI for the knockdown) staining in Figure 8b, which make it difficult to verify the author's conclusion that there is a lack of mature myocardial sarcomere organization in the knockdown.

R: We thank the reviewers suggestion. We have repeated the in vitro induction of cardiomyocyte differentiation using an alternative knockdown strategy (Smart Silencer containing small interfering RNA and antisense oligonucleotides) besides the shRNA knockdown method. The immunofluorescent staining was also repeated, we have replaced the images in Figure 8 with new results. Additionally, we also quantified the *HSALNG0104472*-knockdown effect on cardiomyocyte differentiation efficiency using flow cytometry (Figure 8b).

[Changes in Figure 8: The cell morphology changes during cardiomyocyte differentiation (8a) were replaced for repeated results; quantification of cardiomyocyte differentiation efficiencies has been added (8b); immunofluorescence staining was replaced with newly repeated results]

REVIEWERS' COMMENTS:

Reviewer #4 (Remarks to the Author):

The experimental modification and revised Figure 8 are improvements and responsive to the critique.

Responses point-to-point are as below:

Reviewer #4 (Remarks to the Author):

The experimental modification and revised Figure 8 are improvements and responsive to the critique.

R: We appreciate for the reviewer's valuable comments and consistent attention to our study.